# An improved parameterization of leaf area index (LAI) seasonality in the Canadian

Land Surface Scheme (CLASS) and Canadian Terrestrial Ecosystem Model

(CTEM) modelling framework

<sup>1</sup>Ali Asaadi, <sup>1</sup>Vivek K. Arora, <sup>2</sup>Joe R. Melton, and <sup>2</sup>Paul Bartlett

<sup>1</sup>Canadian Centre for Climate Modelling and Analysis, Environment and Climate Change Canada

<sup>2</sup>Climate Research Division, Environment and Climate Change Canada

Corresponding author: Ali Asaadi, Canadian Centre for Climate Modelling and Analysis, Victoria, BC,

V8W 2Y2, Canada

Email: ali.asaadi@canada.ca

#### Abstract

1 Leaf area index (LAI) and its seasonal dynamics are key determinants of vegetation productivity in nature and as represented in terrestrial biosphere models seeking to understand land-surface atmosphere 2 flux dynamics and its response to climate change. Non-structural carbohydrates (NSCs) and their 3 4 seasonal variability are known to play a crucial role in seasonal variation of leaf phenology and growth and functioning of plants. The carbon stored in NSC pools provides a buffer during times when supply 5 and demand of carbon are asynchronous. An example of this role is illustrated when NSCs from 6 previous years are used to initiate leaf onset at the arrival of favourable weather conditions. In this 7 study, we incorporate NSC pools and associated parameterizations of new processes in the modelling 8 framework of the Canadian Land Surface Scheme-Canadian Terrestrial Ecosystem Model (CLASS-9 CTEM) with an aim to improve the seasonality of simulated LAI. The performance of these new 10 parameterizations is evaluated by comparing simulated LAI and atmosphere-land CO<sub>2</sub> fluxes, to their 11 observation-based estimates, at three sites characterized by broadleaf cold deciduous trees selected 12 from the Fluxnet database. Results show an improvement in leaf onset and offset times with about 2 13 weeks shift towards earlier times during the year in better agreement with observations. These 14 improvements in simulated LAI help to improve the simulated seasonal cycle of gross primary 15 productivity (GPP) and as a result simulated net ecosystem productivity (NEP) as well. 16

- 17
- 18
- 19
- 20
- 21
- 22
- 22
- 23

#### 24 1 Introduction

Biosphere-atmosphere interactions constitute a complex system which plays an important role in the regulation of the climate. These interactions are important determinants governing the physical and 26 27 chemical properties of the atmosphere as well as the growth of plants, and result in the biosphere and 28 atmosphere behaving as a coupled system (Pilegaard et al., 2003). Understanding this coupled behavior is a key research priority due, not only to the important role that terrestrial ecosystems play in 29 30 modulating the global carbon cycle, but also to the significance of land surface characteristics for local 31 and regional climate through biogeophysical effects (Cox et al., 2000; Prentice et al., 2001; Bonan, 2008; Franklin et al., 2016). This growing recognition of the role of land surface vegetation, and its bi-32 directional interactions with the climate system, has led to ever increasing complexity of the physical 33 34 and biogeochemical processes that are incorporated in the land surface components of regional and global climate models (Foley et al., 1996; Sitch et al., 2008; Flato et al., 2013). Process-based land 35 surface schemes and ecophysiological models (e.g., Running et al., 1999; Mäkelä et al., 2000; Friend et 36 37 al., 2007; IPCC, 2013; Sato et al., 2015) simulate atmosphere-land exchanges of carbon, water, and 38 energy, and offer tools for understanding vegetation behaviour for the present climate, and for projecting vegetation behaviour for future climate scenarios. 39

The plant canopy is a locus of physical and biogeochemical processes in an ecosystem. The 41 functional and structural attributes of plant canopies are affected by microclimatic conditions, nutrient 42 dynamics, herbivore activities, and many other factors (Asner et al., 2003). Leaves are the point of contact between plants and atmospheric CO<sub>2</sub>; an increase in leaf area potentially enhances the 43 opportunity for carbon uptake, albeit at the cost of a greater demand for water (Norby et al., 2003). The 44 amount of foliage contained in plant canopies is one of the most basic ecological characteristics that 45 integrates the effects of overall environmental conditions. Canopy leaf area serves as the dominant 46 47 physical control over primary production (photosynthesis), transpiration, energy exchange, and other

physiological attributes pertinent to a range of ecosystem processes, and is therefore a core element of
ecological field and modeling studies (e.g., Knyazikhin et al., 1998; Xavier and Vettorazzi, 2004;
Aboelghar et al., 2010; Gonsamo and Chen, 2014; Bao et al., 2014; Savoy and Mackay, 2015).

LAI (defined as the amount of leaf area  $(m^2)$  in the canopy per unit ground area  $(m^2)$ ) is a 52 dimensionless quantity and therefore can be assessed across a range of spatial scales, from individual plant, a forest stand or grassland, to large regions and continents. Leaf phenology describes the 53 54 response of leaves to seasonal and climatic changes including the timing of bud burst, senescence (leaf 55 maturity or browning), and leaf abscission (leaf fall), and has been documented in a wide range of literature (e.g., Kikuzawa, 1995; Myneni et al., 1997; Arora and Boer, 2005; Menzel et al., 2006; 56 Parmesan, 2006; Richardson et al., 2010; Dragoni et al., 2011; Smith and Hall, 2016). Leaf phenology 57 is a function of environmental conditions (in particular, temperature, soil moisture and day length). The 58 structural and adaptive qualities specific to vegetation type also determine the timing of leaf 59 phenological events. Accurate prediction of recurring vegetation cycles as a function of climate is an 60 61 important feature that vegetation models are expected to reproduce. The timing of bud burst and leaf senescence determine the length of the growing season, and this affects gross and net primary 62 productivities (GPP and NPP), the annual cycle of LAI, and consequently, the energy, water, and 63 carbon fluxes. The seasonal progression of LAI also influences canopy conductance (Blanken and 64 Black, 2004), albedo (Sakai et al., 1997) and through its modulation of sensible and latent heat fluxes 65 66 (Moore et al., 1996) it also affects surface air temperatures (Levis and Bonan, 2004).

Despite its importance, the representation of LAI in terrestrial biosphere models is considered poor (Richardson et al., 2012). Lack of high quality long term observations, the use of prescribed LAI, simplified formulations of underlying biogeochemical processes, and coarse spatial resolution have been mentioned as some of the limitations to accurate representation of LAI (Kucharik et al., 2006). Since canopy seasonality is an important determinant of carbon (C) fluxes, poor representation of the

reasonal dynamics of LAI can lead to inaccurate estimation of vegetation productivity and consequently the net atmosphere-land  $CO_2$  flux (Ryu et al., 2008).

Non-structural carbohydrates (NSCs) are the primary products of photosynthesis and a key energy 75 source for plant growth and metabolism. NSCs play a central role in a plant's life processes and its 76 response to the environmental conditions (Kozlowski, 1992; Ögren, 2000; Chatterton et al., 2006; O'Brien et al., 2014; Hartmann and Trumbore, 2016; Sperling et al., 2017). Previous studies have 77 78 suggested that NSCs are stored in all plant organs (i.e., leaf, branch, root and stem) at different 79 concentrations that vary seasonally and also inter-annualy in response to changes in environmental conditions (e.g., Oberhuber et al., 2011; Bazot et al., 2013; Mei et al., 2015). The amount of NSCs and 80 their particular allocation to leaves, stems, and roots are considered eco-physiological traits and are 81 among the range of adaptive strategies that plants use (Li et al., 2001; Poorter and Kitajima, 2007; 82 Wyka et al., 2016). Many factors influence leaf NSC content, including nutrient elements (Zotz and 83 Richter, 2006), temperature (Gough et al., 2010), precipitation (Würth et al., 2005), drought (Rosas et 84 85 al., 2013), and phenology (Chen et al., 2017). Despite extensive research on the seasonal dynamics of 86 NSC concentrations, the size and relative contributions of NSC pools across different tree organs are not well understood (Mei et al., 2015). 87

Plant NSC stores can compensate for a carbon or nitrogen shortage when current demand surpasses 89 supply due to the seasonality of plant growth, stresses, or disturbances. The seasonal dynamics of NSC concentrations have been studied in various plant species (e.g., Zhu et al., 2012; Richardson et al., 90 91 2013; Saffell et al., 2014). In deciduous plants, when photosynthesis is constrained by limited leaf area 92 and low temperature in early spring, NSC is mobilized from stem and roots to support respiration and 93 tissue growth, resulting in decreased concentrations of NSC in these storage organs (Hoch et al., 2003; Palacio et al., 2007). During the growing season, storage pools are replenished and NSC concentration 94 increases (Teixeira et al. 2007; Klein et al., 2016). Typically, NSC concentrations in storage organs of 95

the short-lived fast-growing species decrease in springtime after bud flush and then increase during the 97 remainder of the growing season. Correspondingly, the storage organs shift from being a NSC source in the early growing season to becoming a sink in the late growing season, maintaining tree survival 98 99 after the termination of photosyntate flow from aboveground sources to supply energy for stem and 100 root tissues through the winter (Würth et al., 2005; Gough et al., 2010). During periods of limited photosynthesis, such as winter dormancy or drought stress, trees depend solely on stored NSCs to 101 102 maintain basic metabolic functions, produce defensive compounds, and retain cell turgor (Sperling et 103 al., 2015). For deciduous species, the NSC storage provides the means to jump start leaf onset by using a part of NSC stores to push leaves out at the onset of favourable weather conditions (e.g. in spring in 104 the northern hemisphere). Representation of NSC pools is therefore an essential step for terrestrial 105 106 biosphere models to better simulate leaf phenology and seasonal variability of LAI.

Here, we include a representation of NSC pools and the associated parameterizations in the framework of the Canadian Land Surface Scheme-Canadian Terrestrial Ecosystem Model (CLASS-108 109 CTEM). CLASS-CTEM exhibits delayed leaf phenology and we attempt to address this issue. In the 110 original model, the simulated global LAI reaches its maximum in August whereas the observed LAI peaks in July (e.g., see Fig. 11 of Anav et al., 2013). The objective of this study is to improve and 111 assess the performance of CLASS-CTEM simulated leaf phenology for broadleaf cold deciduous trees. 112 113 Model performance is evaluated against in situ measurements from three sites from the Fluxnet data network (https://fluxnet.ornl.gov/obtain-data) which provides tower-based meteorological variables 114 used to drive the model as well as observations of LAI, carbon, and energy fluxes. 115

2 Model, data, and methods

#### 119 2.1 CLASS-CTEM model

A coupled version of the Canadian Land Surface Scheme (v. 3.6; Verseghy, 2012) and Canadian 121 Terrestrial Ecosystem Model (v. 2.1.1, Melton and Arora 2014) (CLASS-CTEM) is used here. Slightly 122 older versions of both models are currently implemented in the second generation Canadian Earth 123 System Model (CanESM2; Arora et al., 2011). While CLASS simulates fluxes of energy, water, and momentum at the land-atmosphere boundary, atmosphere-land fluxes of CO<sub>2</sub> are simulated by CTEM. 124 125 CLASS operates at a typical time step of 30 minutes and prognostically simulates the liquid and frozen 126 soil moisture and soil temperature for its multiple soil layers (3 layers are employed here with maximum thicknesses of 0.1, 0.25 and 3.75 m); the temperature, thickness and fractional cover of 127 snow; and the temperature and amount of snow and rain on the vegetation canopy. The permeable 128 129 depth of the soil column may be smaller than the total depth of the soil layers employed; if a layer spans the permeable depth boundary it is subdivided for hydrological calculations. CLASS 130 131 distinguishes four plant functional types (PFTs) (needleleaf trees, broadleaf trees, crops, and grasses) as 132 shown in Table 1. CLASS calculates net radiation  $(R_n)$  based on prognostically calculated land surface 133 albedo and the skin temperature of the land surface  $(T_s)$  as

$$R_n = SW(1-\alpha) + LW - \sigma T_s^4 \tag{1}$$

where  $\alpha$  is albedo, SW and LW are incoming short and long wave radiation,  $\sigma$  is the Stefan-Boltzman constant.  $R_n$  is partitioned into latent (LE), sensible (H), ground, and canopy heat fluxes. When in equilibrium and over annual and longer time periods, since ground or canopy do not gain or lose heat systematically, the sum of latent and sensible heat fluxes equals net radiation ( $R_n = LE + H$ ).

CTEM simulates terrestrial processes by prognostically tracking carbon in three living vegetation 140 components (leaves, stems and roots) and two dead carbon pools (litter and soil) for seven non-crop 141 and two crop PFTs that map directly onto CLASS' PFTs (Table 1). The terrestrial ecosystem processes

simulated in this study include photosynthesis, autotrophic respiration, heterotrophic respiration, dynamic leaf phenology, and allocation of carbon from leaves to stem and root components. These processes are described in a sequence of papers detailing parameterization of photosynthesis, autotrophic and heterotrophic respiration (Arora, 2003), dynamic root distribution (Arora and Boer, 2003), phenology, carbon allocation, biomass turnover and conversion of biomass to structural attributes (Arora and Boer, 2005). A full description of CTEM can be found in the appendix of Melton and Arora (2016).

The structure of CTEM is shown in Fig. 1; the original three live vegetation pools (leaves, stem, and roots) are indicated by L, S, and R subscripts, respectively), and the two dead carbon pools (litter or detritus and soil carbon) are indicated by D and H subscripts, respectively). Time varying fluxes in and out of these carbon pools ( $C_L$ ,  $C_S$ ,  $C_R$ ,  $C_D$ , and  $C_H$ ; in kgCm<sup>-2</sup>) makes them prognostic variables in the model. The corresponding rate change equations for amount of carbon in the three live vegetation components (leaves, stem, and roots) in the original model version are represented by

$$\frac{dC_i}{dt} = a_{f_i} (G - E_m - E_g) - D_i = a_{f_i} N - D_i$$
(2)

where the index *i* corresponds to each of the live vegetation pools (i = L, S, R),  $a_{f_i}$  represents allocation 156 fraction for a given vegetation component, G is canopy gross primary productivity,  $E_m$  is vegetation 157 maintenance respiration,  $E_g$  is vegetation growth respiration,  $D_i$  represents the litter loss. N = G - G158  $E_m - E_g$  is the canopy net primary productivity (NPP) and therefore  $a_{f_i}N$  represents fraction of NPP 159 allocated to the three vegetation components. Growth respiration,  $E_{g}$ , is estimated as a fraction of the 160 161 positive gross canopy photosynthetic rate after maintenance respiration has been accounted for (equation A28, Melton and Arora (2016)).  $E_a = E_m + E_g$  is the autotrophic respiration, therefore, N =162 163  $G - E_a$ . When heterotrophic respiration  $(E_h)$  is accounted for, net ecosystem productivity (NEP) is calculated as  $NEP = G - E_a - E_h = N - E_h$ . Positive NEP values indicate that land is gaining carbon 164

from the atmosphere. Combined autotrophic and heterotrophic respiration  $(E_a + E_h)$  are referred to as the ecosystem respiration  $(E_r)$ .

## 168 2.1.1 Addition of NSC pools

For the modifications made in this study, first, NSC pools are included in each of the live vegetation components (leaves, stem, and roots). The total biomass (Kg C m<sup>-2</sup>) for each of these 170 171 components is divided into its non-structural and structural components (indicated by subscripts NS 172 and S) as shown in Figure 1.  $C_L = C_{L,NS} + C_{L,S}$  and similarly for  $C_S$  and  $C_R$ . The fraction of NPP allocated to each live vegetation component is first moved to its non-structural part, and a flux of 173 174 carbon from the non-structural to the structural part provides carbon to the structural part. Once the 175 carbon is moved from non-structural to a structural part of a component it cannot be moved back. Since NPP includes respiratory losses, this essentially implies that respiratory carbon losses are assumed to 176 177 occur from the non-structural part. Litter losses, on the other hand, occur from both the structural and non-structural parts of leaves, stem and root components. 178

The modified rate change equations for carbon in the non-structural and structural parts of leaf (Eq.

3) and stem and root (Eq. 4) components are thus written as

$$\frac{dC_{L,NS}}{dt} = a_{f_L}N - D_{L,NS} - F_{ns2s,L} + T_S + T_R 
\frac{dC_{L,S}}{dt} = F_{ns2s,L} - D_{L,S} 
\quad \frac{dC_{j,NS}}{dt} = a_{f_i}N - D_{j,NS} - F_{ns2s,j} - T_j
\frac{dC_{j,S}}{dt} = F_{ns2s,j} - D_{j,S} 
(3)$$

where  $F_{ns2s,i}$  (i = L, S, R) represents carbon flux from the non-structural to structural part of a component (leaf, stem or root), and  $T_j$  (j = S, R) represents the reallocation (or transfer) of carbon from stem and root components to leaves during leaf out period. Note that there are no autotrophic respiration terms in equations (3) and (4) since they are already included in the term *N*, the net primary productivity.  $F_{ns2s,i}$  is represented as

$$F_{ns2s,i} = \mu_i a_{f_i} Nmax[0, (\eta - \eta_{i,min})]$$
 (5)

where  $\mu_i$  is a non-dimensional coefficient set to 70. Equation (5) attempts to keep the fraction of 191 192 non-structural to total carbon in a component  $\eta_i = C_{i,NS}/C_i$  above its minimum specified value  $\eta_{i,min}$ . During periods of negative NPP, for e.g. as is the case during winter for cold deciduous trees when they 193 194 do not have their leaves on,  $F_{ns2s,i}$  is set to zero. This represents the attempts by plants to conserve their 195 NSC pools during a period of no productivity. The amount of carbon in non-structural and structural parts of all vegetation components are time varying variables and therefore so is the ratio of non-196 197 structural to total carbon  $(\eta_i)$ . The minimum ratio of non-structural to total carbon in a component  $(\eta_{i,min})$  is specified to be 0.05 for the broadleaf cold deciduous PFT considered here, following Li et al. 198 199 (2016).

The above modifications made to version 2.1.1 of CTEM in regards to the inclusion of NSC pools 200 201 allow the movement of non-structural carbohydrates between the model's three live vegetation 202 components, in particular, reallocation of non-structural carbohydrates from stem and root components for leaf out at the onset of spring for the broadleaf cold deciduous tree PFT. In addition, we also adjust 203 204 allocation fractions for the leaves, stem and root components after summer solstice in response to day 205 length, and the lower temperature thresholds for leaf litter generation due to cold stress. Deciduousness at high latitudes is determined both by day length and temperature (Xie et al., 2015) and these 206 207 modifications, discussed below, help to improve simulated leaf phenology.

# 208

### 209 2.1.2 Reallocation of non-structural carbon during leaf out period

Leaf phenology in CTEM is represented via four phenological states a plant can be in at any given 211 time (Arora and Boer, 2005). These stages include no leaves or dormant state, maximum leaf growth 212 state, normal growth state, and leaf fall or harvest state. Depending on their deciduousness, CTEM's nine plant functional types (Table 1) may or may not go through these four different leaf phenological 213 214 states. A broadleaf cold deciduous tree, transitions through all four states in a year: leafless/dormant 215 state in winter, maximum growth state (following arrival of favorable climatic condition in spring when all NPP is allocated to leaves), normal growth state (after reaching a threshold LAI, NPP is allocated to 216 stem and roots in addition to leaves), and finally the leaf fall state (triggered by unfavorable 217 218 environmental conditions and with no carbon allocation to leaves). When all the leaves have been shed, the trees go back into the leafless or dormant state again and the cycle repeats itself in the next year. 219

In the original version of the model, when a plant moves into the maximum leaf growth state all NPP is allocated to leaves until a threshold LAI ( $L_{thrs}$ , m<sup>2</sup>/m<sup>2</sup>) has been grown.  $L_{thrs}$  is about 40%-221 222 50% of the maximum LAI a plant can support depending on its stem and root biomass and based on an allometric relationship between green and woody biomass (Melton and Arora, 2016). In the absence of 223 224 NSC pools in the original model version, photosynthesis during the early leaf out period is based on a 225 small imaginary amount of leaves (referred to as storage LAI). Once the actual LAI exceeds the storage 226 LAI then photosynthesis is based on the actual LAI. Storage LAI is proportional to a plant's stem and root biomass and was intended as a proxy for the size of NSC pools. However, the rate of 227 photosynthesis from a reasonably apportioned storage LAI is still too small to realistically 'push out' 228 229 leaves at the onset of spring in a time period of about two weeks as seen in observations. This 230 limitation in the original model version is overcome here with the inclusion of NSC pools. In the modified version of the model used here, a specified fraction of carbon amount needed to reach the 231

threshold LAI is reallocated ( $T_j$ , j = S, R) from a plant's stem and root NSC pools to the non-structural part of leaves every day until LAI reaches  $L_{thrs}$ . Note that while this reallocation occurs the leaves are still able to photosynthesize and able to increase their biomass as in the original model version, depending on meteorological conditions. The objective of carbon reallocation from stem and roots to leaves is to accelerate the rate of leaf expansion and LAI increase during leaf onset.

The amount of carbon reallocated (kg  $C/m^2$ ) from stem and root components to leaves is given by

238 
$$T_j = \beta \frac{L_{thrs}}{SLA} f_j; j = S, R$$
(6)

239 
$$f_j = \begin{cases} \frac{c_{j,NS}}{c_{S,NS}+c_{R,NS}} & if\eta_j > \eta_{j,min} \\ 0 & if\eta_j \le \eta_{j,min} \end{cases}; j = S, R$$

$$(7)$$

240

where SLA is the specific leaf area (m<sup>2</sup>/kg C),  $\beta$  is the reallocation coefficient set to 6.66x10<sup>-3</sup> and fractions  $f_j(j = S, R)$  ensure that carbon reallocated from stem and root NSC pools is proportional to the size of their NSC pools. Equation (7) also shows that when the fraction of NSC pool relative to total carbon in a component ( $\eta_j = C_{j,NS}/C_j$ ), j = S, R is equal to or drops below its minimum specific value ( $\eta_{j,min}$ ) then reallocation is stopped. Reallocation is only performed during the leaf out state when trees are in the maximum leaf growth state.

247

# 248 2.1.3 Adjustments to allocation fraction to leaves after the summer solstice

CTEM uses dynamically calculated allocation fractions (Arora and Boer, 2005; Melton and Arora, 2016) for leaves, stem, and roots, which are based on the light, water, and leaf phenological status of vegetation. The allocation to the three live vegetation components is based on assumptions that carbon is preferentially allocated: 1) to roots when soil moisture is limiting, 2) to leaves when LAI is low, and 3) to stem to increase vegetation height and lateral spread when increasing LAI leads to a decrease in

light penetration. These allocation fractions are superseded by three additional rules: 1) all carbon is
allocated to leaves at the time of leaf out for cold deciduous tree PFTs to accelerate leaf development,
allocation fractions are adjusted when necessary to ensure a tree has enough stem and root biomass
to support leaves (to satisfy a structural allometric relationship), and 3) a minimum realistic root to
shoot ratio is maintained for all PFTs.

When compared to observation-based estimates of globally-averaged LAI, CLASS-CTEM simulated LAI shows a much slower rate of decline after reaching its annual maximum, which typically occurs just after the summer solstice in each hemisphere (Anav et al., 2013). To address this issue allocation to leaves of cold deciduous tree PFTs after summer solstice is reduced by multiplication with a day-length dependent factor ( $\Gamma$ ) given by

264 
$$\Gamma = \left[\frac{d}{d + (d_{max} - d)0.5(tanh(\frac{\pi}{180}(20\phi - 800)) + 1)}\right]^{20}$$
(8)

265 
$$d = 24 - \frac{24}{\pi} a \cos\left[max\left(-1, min\left(\frac{\sin\phi\sin\delta_c}{\cos\phi\cos\delta_c}, 1\right)\right)\right]$$
(9)

266

where d is the day length at latitude  $\phi$  (radian), d<sub>max</sub> is its maximum value (hour), and  $\delta_c$  (radians) is 267 268 solar declination.  $\Gamma$  varies between 0 and 1 and its behaviour in Figure 2 shows how allocation to 269 leaves is reduced at a faster (slower) rate closer to poles (equator) after summer solstice in the northern hemisphere (June 21). Below 30°N in the northern hemisphere equation (8) yields  $\Gamma = 1$  so allocation 270 fraction for leaves is not modified. Deciduousness due to day length and temperature typically does not 271 occur in tropics where it is primarily controlled by soil moisture. Neither do broadleaf deciduous cold 272 trees typically exist in the tropics. Similar behaviour is obtained for the southern hemisphere after 273 December 21. Since the allocation fractions for leaves, stem, and root components should add to 1 the 274