# Peer review of "An improved parameterization of leaf area index (LAI) seasonality in the Canadian"

_Biogeosciences, 2018_

## Referee Comment (RC1) · Anonymous Referee #1 · 21 May 2018

Dear Authors,

These are my comments of the work entitled: An improved parameterization of leaf area index (LAI) seasonality in the Canadian Land Surface Scheme (CLASS) and Canadian Terrestrial Ecosystem Model (CTEM) modelling framework. Firstly, I want to thank the authors for the work done. I felt the manuscript quite interesting.

This paper describes the addition of the Non Structural Carbohidrates (NSC) module,

to the CLASS-CTEM model. NSC module allows to better represent Leaf Area seasonality, as well as to provide a mobile carbohydrate pool to the trees to increase its resilience to disturbances in absence of photosynthesis. It is tested in three Fluxnet sites, where GPP, LAI, and heat fluxes (Incident radiation, latent heat and sensible heat) model projections are contrasted against real data. In my opinion, this is an interesting, thoroughfull work, where the authors clearly demonstrate that the addition of the NSC module clearly improves model performance. My major concerns about the present paper are about its novelty. Currently most of process-based forest simulation models does include the NSC module (Fontes et al., 2010), in a similar way than the new module for the CLASS-CTEM model. So, in my opinion, your current manuscript doesn't clarify the novelty of your work. Furthermore, throughout your manuscript there is little reference to other models that include this key compartment, and I think it would be a nice element to include in the discussion, as there is plenty of other works in which the addition of NSC in a given model clearly improves its performance.

Specific considerations:

- I felt a little lacking how the Maintenance respiration was calculated in CLASS-CTEM. I've seen other reviewers asking for the same point, and I feel like an addition of the maintenance respiration formulae as well as the assumptions of the model about this process would improve significantly the paper. Besides, I have a couple of questions about maintenance respiration: it is dependent on temperature? It is assumed the same respiration rate for the structural and non-structural carbohydrates?

- It is a minor issue, but, in general, I think your explanations about Leaf Area (LA) importance upon photosynthesis. However, I think you are wrong when referring to them as LAI (for example, lines 1, 63). LAI doesn't perform photosynthesis, it is the Leaf Area, that does it. LAI is just an explanatory index about the surface of leaf area per unit of surface.

- In lines 175-177, you state that respiratory carbon loses are assumed to occur from

the non-structural part. Does it mean that structural carbon is not accounted in maintenance respiration? I guess you did not mean that, but as it is stated, it may lead to misinterpretations.

- In point 2.1.1 (Reallocation of non-structural carbon during leaf out period), you do state that "after reaching a threshold LAI, NPP is allocated to stem and roots in addition to leaves". I could not find in your work how these compartments are developed. Do your model follow any predefined rule (e.g. the Pipe model rule, Shinozaky, 1974)? Or they are equally allocated throughout the tree compartments according to a predefined rate?

- Line 372-373. I would remove the sentence "The figure legends, in addition to identifying the two mode versions and observations, also show the mean annual value of the quantity plotted", and I would include it in the figure footnote.

- Lines 419-420. Your sentence assumes an equilibrium between the atmospheric CO2 and the biosphere. Maybe is far beyond the discussion of your paper, but I think that this is not strictly true, as there has been previous works identifying the instability of the atmosphere-biosphere complex (e.g. Higgins et al., 2002). It is a minor change, but I would suggest to erase the "currently" in the sentence, thus indicating the responsive nature of biosphere to historical changes in atmospheric CO2.

- Line 451. Again, following which rule, besides the "after reaching a LAI threshold", are the carbohydrates allocated through the three compartments? A fixed rate? A mechanistic rule?

- Point 3.3. Here, you find that you do overestimate latent heat when modelling your three forests. Are there any research papers about evapotranspiration experiments in those forests? If they are, maybe you should transform latent heat into evapotranspiration values, so you can compare them to your data, and you might then have a better explanation about why does your model overestimates so high the evapotranspiration (Latent heat). In addition, how do this relate to your overestimation of Leaf Area? I

see a little discussion about it in lines 508-512, but I think you minimize the effect that you overestimate the LAI during the growing season by about 2 m2/m-2 in each plot, and this affect both the evapotranspiration at canopy level, but also to the evaporative energy available at ground level.

- Lines 519 to 523. Please, revise the Vmax concept: as states the original paper from Kattge et al, (2009), Vmax is the maximum carboxilation capacity, not the maximum photosynthetic rate. In addition: you are justifying a tautology: This is the parameter that we want to apply to Vmax, so we adjust all other model parameters to fit the results according to this Vmax value. In addition, you finish the sentence with "it is possible that the average Vmax value derived by Kattge et al. (2009) is not representative of [...]". I agree that mean Vmax value not representing correctly your forest performance is a possible explanation, but I would rather discuss that Vmax is not the only constrain to photosynthesis, as Jmax is also limiting assimilation rate.

- Figures 4-6. I would expand a little the footnote, to include the information that results are represented as averaged daily values. In addition, I would consider to change the Leaf Area Index inner pannel to represent the median values during the vegetative period rather than average values, as I think they would be more indicative of the similarities-differences between Fluxnet measurements and model outputs.

---

## Author Comment (AC1) · 19 Jun 2018

We thank the reviewer for taking the time to review our manuscript and her/his constructive comments. Our point-by-point responses are included below. The reviewer's comments are indicated in italic font and our responses in regular font.

*Reviewer:*

*These are my comments of the work entitled: An improved parameterization of leaf*

[Figure]

*area index (LAI) seasonality in the Canadian Land Surface Scheme (CLASS) and Canadian Terrestrial Ecosystem Model (CTEM) modelling framework. Firstly, I want to thank the authors for the work done. I felt the manuscript quite interesting. This paper describes the addition of the Non Structural Carbohydrates (NSC) module, to the CLASS-CTEM model. NSC module allows to better represent Leaf Area seasonality, as well as to provide a mobile carbohydrate pool to the trees to increase its resilience to disturbances in absence of photosynthesis. It is tested in three Fluxnet sites, where GPP, LAI, and heat fluxes (Incident radiation, latent heat and sensible heat) model projections are contrasted against real data. In my opinion, this is an interesting, thoroughfull work, where the authors clearly demonstrate that the addition of the NSC module clearly improves model performance. My major concerns about the present paper are about its novelty. Currently most of process-based forest simulation models does include the NSC module (Fontes et al., 2010), in a similar way than the new module for the CLASS-CTEM model. So, in my opinion, your current manuscript doesn't clarify the novelty of your work. Furthermore, throughout your manuscript there is little reference to other models that include this key compartment, and I think it would be a nice element to include in the discussion, as there is plenty of other works in which the addition of NSC in a given model clearly improves its performance.*

**Authors:**

We will modify our manuscript to include references to existing work on inclusion of NSC pools in ecosystem models. Although NSC have been included in other process-based models, the CLASS-CTEM model lacked this part and the present study is the first effort to address this problem. While we do not claim novelty for the present study, we will clarify and stress the importance of NSC pools even more when revising our manuscript.

***Reviewer:***

*Specific considerations:*
*I felt a little lacking how the Maintenance respiration was calculated in CLASS-CTEM.*

*I've seen other reviewers asking for the same point, and I feel like an addition of the maintenance respiration formula as well as the assumptions of the model about this process would improve significantly the paper. Besides, I have a couple of questions about maintenance respiration: it is dependent on temperature? It is assumed the same respiration rate for the structural and non-structural carbohydrates?*

*In lines 175-177, you state that respiratory carbon loses are assumed to occur from the non-structural part. Does it mean that structural carbon is not accounted in maintenance respiration? I guess you did not mean that, but as it is stated, it may lead to misinterpretations.*

**Authors:**

The text was modified and some clarifications were added in response to the editor's review (lines 160-162 and 187-189) in the context of maintenance respiration. Although the model's maintenance respiration along with other processes have already been discussed in previous publications (e.g., section A3.1 in Melton and Arora, 2016), we will add more complementary text and equations to show how the model calculates maintenance respiration . Yes, indeed maintenance respiration is temperature dependent. In the modified (i.e., NSC-added) version of the model presented in this study, maintenance respiration occurs only from the non-structural pools consistent with observational and modelling studies (e.g., Hoch et al., 2003; Sperling et al., 2015; and Li et al., 2016) which show that plants' NSC stores become depleted during dormant season and cold and drought stresses due to excess respiration.

*Reviewer:*

*It is a minor issue, but, in general, I think your explanations about Leaf Area (LA) importance upon photosynthesis. However, I think you are wrong when referring to them as LAI (for example, lines 1, 63). LAI doesn't perform photosynthesis, it is the Leaf Area, that does it. LAI is just an explanatory index about the surface of leaf area per unit of surface.*

**Authors:**

We feel this is a terminology issue. While it is true that photosynthesis is a function of the total "leaf area", several of the model's parameterizations use leaf area index. For example, CLASS-CTEM model uses the Beer-Lambert law ($1 - e^{-k(LAI)}$, where k is a vegetation-dependent light extinction coefficient) to scale photosynthesis from the leaf to the canopy level. Nevertheless, we will try to clarify this terminology issue.

***Reviewer:***
*In point 2.1.1 (Reallocation of non-structural carbon during leaf out period), you do state that "after reaching a threshold LAI, NPP is allocated to stem and roots in addition to leaves". I could not find in your work how these compartments are developed. Do your model follow any predefined rule (e.g. the Pipe model rule, Shinozaky, 1974)? Or they are equally allocated throughout the tree compartments according to a predefined rate?*
*Line 451. Again, following which rule, besides the "after reaching a LAI threshold", are the carbohydrates allocated through the three compartments? A fixed rate? A mechanistic rule?*

**Authors:**
While the model's several processes have been described before in Melton and Arora (2016), we will add additional details to clarify these allocation-related questions raised by the reviewer. In brief, the model uses dynamically calculated allocation fractions for leaves, stems and roots based on light, water and leaf phenological status of vegetation. The preferential allocation of carbon to the different tissue pools is based on three assumptions: (i) if soil moisture is limiting, carbon should be preferentially allocated to roots for greater access to water, (ii) if LAI is low, carbon should be allocated to leaves for enhanced photosynthesis and finally (iii) carbon is allocated to the stem to increase vegetation height and lateral spread of vegetation when the increase in LAI results in a decrease in light penetration. The dynamically calculated allocation fractions are further modified to ensure that root to shoot ratio is realistic and that there is enough woody biomass to support green biomass.

*Reviewer:*
*Line 372-373. I would remove the sentence "The figure legends, in addition to identifying the two mode versions and observations, also show the mean annual value of the quantity plotted", and I would include it in the figure footnote.*

**Authors:**
Agreed.

*Reviewer:*
*Lines 419-420. Your sentence assumes an equilibrium between the atmospheric $CO_2$ and the biosphere. Maybe is far beyond the discussion of your paper, but I think that this is not strictly true, as there has been previous works identifying the instability of the atmosphere-biosphere complex (e.g. Higgins et al., 2002). It is a minor change, but I would suggest to erase the "currently" in the sentence, thus indicating the responsive nature of biosphere to historical changes in atmospheric $CO_2$.*

**Authors:**
Agreed.

*Reviewer:*
*Point 3.3. Here, you find that you do overestimate latent heat when modelling your three forests. Are there any research papers about evapotranspiration experiments in those forests? If they are, maybe you should transform latent heat into evapotranspiration values, so you can compare them to your data, and you might then have a better explanation about why does your model overestimates so high the evapotranspiration (Latent heat). In addition, how do this relate to your overestimation of Leaf Area? I see a little discussion about it in lines 508-512, but I think you minimize the effect that you overestimate the LAI during the growing season by about 2 $m^2/m^2$ in each plot, and this affect both the evapotranspiration at canopy level, but also to the evaporative energy available at ground level.*

**Authors:**

As the reviewer noted evapotranspiration and latent heat fluxes are the same quantity but in different units. Evapotranspiration is expressed in water units ($mm$) and latent heat flux is expressed in energy units ($W/m^2$). At the flux net sites latent heat flux is available but as mentioned in section 3.3 of the manuscript it suffers from energy balance closure. So it is difficult to conclusively determine how much of this overestimation is due to overestimation by the model and how much of it is due to bias in the observed data. Our experience with the CLASS-CTEM model in the past has been that in locations where soil moisture constraint is not very large, as is the case at these three sites, total evapotranspiration is controlled by available energy. Hence the expected seasonality in latent heat flux is characterized by higher values during summer and lower during winter at these sites. Since the latent heat flux at these three sites is primarily controlled by available energy the resulting implication is that if evaporative demand cannot be met by transpiration then it will be met by evaporation from the soil. As a result, changes in LAI do not significantly affect total evapotranspiration (or latent heat flux) but change the partitioning of evapotranspiration flux coming from transpiration, evaporation of intercepted water on canopy leaves, and evaporation from the soil. Therefore, had the model simulated lower LAI than it currently does, then the latent heat flux would not have been significantly different from its current values. We will include additional discussion along these lines when revising our manuscript.

***Reviewer:***
*Lines 519 to 523. Please, revise the Vmax concept: as states the original paper from Kattge et al, (2009), Vmax is the maximum carboxilation capacity, not the maximum photosynthetic rate. In addition: you are justifying a tautology: This is the parameter that we want to apply to Vmax, so we adjust all other model parameters to fit the results according to this Vmax value. In addition, you finish the sentence with "it is possible that the average Vmax value derived by Kattge et al. (2009) is not representative of [...]". I agree that mean Vmax value not representing correctly your forest performance is a possible explanation, but I would rather discuss that Vmax is not the only constrain to photosynthesis, as Jmax is also limiting assimilation rate.*

**Authors:**
Yes, indeed Vmax is the maximum carboxilation capacity. Thank you for catching this. The CLASS-CTEM model uses the Farquhar approach for modelling photosynthesis, and therefore photosynthesis is limited by carboxilation capacity but also light and transport capacity limited rates. However, Vmax remains a very strong parameter that controls photosynthesis in the model. We will clarify this aspect when revising our manuscript. We did not adjust other model parameters to accommodate Kattge et al. (2009) Vmax values. In fact, the very first version of the model used a value of Vmax of 65 $umolCO_2/m^2s$ for its broadleaf cold deciduous PFT which is close to the value of 57 $umolCO_2/m^2s$ suggested by Kattge et al. (2009).

It is probably worth mentioning that at the global scale the performance of CLASS-CTEM model when implemented in the Canadian Earth System Model (CanESM2) is fairly realistic. Figure 11 of Anav et al. (2013), who compared different Earth system models used in the phase 5 of the coupled model intercomparison project (CMIP5), shows that the CLASS-CTEM's simulated "annual global mean" LAI is among one of the best estimates compared to other models. Figure 9 of Anav et al. (2013) shows that the model also does a good job in simulating global annual GPP.

*Reviewer:*
*Figures 4-6. I would expand a little the footnote, to include the information that results are represented as averaged daily values. In addition, I would consider to change the Leaf Area Index inner pannel to represent the median values during the vegetative period rather than average values, as I think they would be more indicative of the similarities-differences between Fluxnet measurements and model outputs.*

**Authors:**
Thank you for pointing this out. We will change the annual mean LAI to mean LAI over the growing season. We will also modify figures' footnotes.

[Figure]

**References:**

Anav, A., and Coauthors, 2013: Evaluating the land and ocean components of the global carbon cycle in the CMIP5 earth system models. J. Climate, 26, 6801–6843.

Hoch, G., A. Richter, C. Körner, 2003: Non-structural carbon compounds in temperate forest trees. Plant Cell Environ., 26, 1067–1081.

Kattge, J., W. Knorr, T. Raddatz, and C. Wirth, 2009: Quantifying photosynthetic capacity and its relationship to leaf nitrogen content for global-scale terrestrial biosphere models. Global Change Biology, 15, 976–991.

Li, N., N. He, G. Yu, Q. Wang, J. Sun, 2016: Leaf non-structural carbohydrates regulated by plant functional groups and climate: evidences from a tropical to cold-temperate forest transect. Ecological Indicators, 62, 22–31.

Melton, J. R., V. K. Arora, 2016: Competition between plant functional types in the Canadian Terrestrial Ecosystem Model (CTEM) v. 2.0, Geosci. Model Dev., 9, 323–361.

Sperling, O., J. M. Earles, F. Secchi, J. Godfrey, M. A. Zwieniecki, 2015: Frost induces respiration and accelerates carbon depletion in trees. PLoS One, 10:e0144124.

---

## Referee Comment (RC2) · Anonymous Referee #2 · 26 Jun 2018

General Comments

The study by Asaadi et al. aims to improve the seasonal timing of LAI simulated by the CTEM model by including a representation of non-structural carbon (NSC) pools and fluxes in the model (in addition to a few other modifications). The new developments in the model are tested at three temperate broadleaved deciduous sites against LAI, carbon and energy flux observations. They show an improvement in the timing of

various stages in the phenological cycle, and corresponding improvement in the timing of carbon fluxes (though limited improvement in energy fluxes).

This is of use to the land surface modeling community as not all LSMs currently include specific NSC pools and related processes. As the authors have discussed, NSC processes are relevant for other components of biogeochemical cycling or ecosystem functioning, in addition to phenology that they focus on in this study (though this discussion could be expanded).

The paper is clearly written and structured (although the goals of their model development work might be better stated as questions or hypotheses). However, I have some concerns about the lack of breadth of the study and depth of the analyses undertaken, which are detailed more in specific comments and outlined briefly here.

1) The addition of non-structural carbon pools and associated fluxes is the primary focus of the study; however, there are no observations of NSCs used to evaluate the authors' modifications to the model. I would have expected that any model modification would be tested against observations that are directly relevant to the new processes added in the model even though I appreciate that NSC data are scarce (see Dietze et al., 2014 for a review of previous such studies in the literature). Why was this not the case? Although the authors state the sites chosen were those with available LAI data, was it not possible to evaluate this model at any site that had observations of non-structural carbon pools (even if those data came from sites that did not also have LAI data)? The authors only chose three sites representing only one plant functional type. I would think there are a greater number of sites with LAI data that this model could be tested against.

2) While the authors detail improvements in their modified model in comparison with observations (though less so for energy fluxes), it is not clear which of the model modifications made (detailed in Sections 2.1.1 to 2.1.4) are responsible for the improvements in the simulated carbon and energy fluxes. The authors could show the impact of each

modification individually, before evaluating all together. I think the modeling community would appreciate knowing how each of the different modifications made to the model contributed to the overall improvement in the model. Such an analysis would help them ascertain if there are potential structural deficiencies in their own model, thus placing this work in a wider context.

3) The analysis lacks depth – namely, there is a lack of a rigorous quantitative evaluation of the modified model. It would be useful to include certain metrics to quantify the improvements simulated by the modified model (simple correlations for example). In addition, given the authors state that their primary goal is to address the issue of delayed leaf phenology, their analyses should be focused only on that question; general discussions of model behavior and magnitude of fluxes are distracting, especially given they have decided not to run a historical or transient simulation of the model after the spinup, with increasing CO2 and climate.

4) The authors state in the discussion around lines 534-535 that the omission of NSC pools in the original model was a structural error. However, they do not definitively provide evidence to support their claim that the omission of NSC pools was a structural error. While their results show that this process can improve model LAI temporal dynamics, they have not conclusively shown that this is the only process that could be responsible for any discrepancies between the model and the observations, and therefore how important it is to add these specific processes. Incorporation of NSC pools and fluxes may not be the only process that can alleviate the problems in the simulated LAI. As they go on to state, biological systems are complex and difficult to represent with physical equations in models. To ensure that we do have the right model behavior, the processes we include must be rigorously tested against data corresponding to that process. Ideally, the authors would test alternative functions available in the literature for the processes they have implemented, in order to estimate the structural uncertainty associated with the new model developments. A Bayesian model selection framework could be used in order to select the most parsimonious model based on a

model selection criterion (such as the Akaike Information Criterion – see Melaas et al., 2013 for an example). I would also be interested to see an analysis on the uncertainty related the parameters they have implemented. It might then be useful to discuss other NSC related processes that remain poorly understood that are not captured by their new model.

5) The discussion lacks depth as to how the models they have implemented compare to other studies that have already implemented NSC models, as well as a discussion of any caveats to their modeling work related to the points I mention here. See specific comments.

Specific Comments

Introduction

Lines 109-111: Unless I have misunderstood, this model has been used in a phenology comparison at these sites (Richardson et al., 2012). If I have the right model, it seems to me that the problems in the behavior of CTEM (for simulating LAI) shown in Richardson et al. are different to that in Anav. This shows that there might be other issues in the phenology models already implemented in CTEM due to differences between versions/parameterizations, without the addition of new processes/modifications to the model?

Model, data and methods

Sections 2.1.1 and 2.1.2 There is a lack of references and/or reasoning for some of the mechanisms they are implementing in the models and the various assumptions they make in doing so (e.g. assumption that respiratory losses occur from non-structural part – line 176, and the references/reasoning behind formulation in equations 5, 6 ).

In addition, the reasoning behind fixing certain parameter values needs to be detailed (e.g. why ðÍÌJǦ = 70 line 191; beta in line 241). Were the parameter values found from the literature, or perhaps they were calibrated based on sensitivity studies or

optimization experiments?

Line 186: maybe refer to Section 2.1.2 for how is Tj calculated?

Section 2.1.3 Are you referring to the fact that CLASS-CTEM has a flat peak of around 2 months in Anav et al., 2013 Fig 11, as opposed to the sharper (∼1 month) peak seen in the observations and other models? In that case, it might be good to just state this in parentheses, as I was distracted by the fact that LAI simulated by CLASS-CTEM does not start to decline until long after the summer solstice (Anav et al., 2013 Fig 11) and much later than the observations. In any case, is there evidence in the literature that allocation fractions is modulated by day length – as represented in Section 2.1.3? No references are given to support the addition of this process. Could this slower rate of decline be due to incorrect parameters/processes related to senescence? Initially I was more distracted by the fact the seasonal cycle is delayed (out of phase) by a month or so. I am not therefore convinced if this correction factor based on day length presented in Section 2.1.3 is needed on top of other structural changes in the model.

Section 2.1.4 Similarly to Sections 2.1.1 (above), why is a value of 12°C now used for Tleaf_cold? Is this based on the literature, or experiments, or a calibration exercise? Please give details and/or references as to how this value was chosen.

Section 2.2.2 Please could you detail where you got the site meteorological data from, and which method and/or software you used to gap-fill the met data? Also, please could you detail why you chose to use a $CO_2$ concentration of 350ppm (this is detailed around line 418 in the results, but needs to be put here). Finally, please could you detail how the LAI measurements were made at each site? Are there differences between sites? This information would be helpful for readers.

Results

Figures 4-6: It would be good to state that both simulated and observed values represent averaged daily values across all years where data are present in the figure

captions.

Section 3.1 It would be helpful to have some metrics that show improvement (or lack thereof) between model versions for the full timeseries at each site. Even just RMSE or R would be helpful to quantify this and help put the results in context. This could be added to Table 3 for example.

Lines 382-384. It would be helpful if the authors showed a comparison of the observations and the model for each of the different modifications to the model that the authors have made in this study (as described in Sections 2.1.1-2.1.4), in addition to the overall improvement brought about by all modifications together. That way, other modeling groups can assess which modifications might be necessary for their own model – thus making the study useful in a wider context. These may only be put supplementary figures or tables, but it would still be useful to discuss in the text.

Lines 390-391: This is true, and a good result. However, I also noted there seems to be an offset in the start of LAI and GPP in the observations at both US-MMS and US-UMB. At US-MMS the onset of LAI now better matches the observations, but there is now a bias towards a too early rise in GPP. Similarly, although the LAI at US-UMB now better matches the observations, and the GPP matches the observations very well, there is a still this offset. Why do you think that is? Is there a discrepancy between the two types of observations?

Lines 396-414: Why haven't the authors run a historical simulation after their spin-up using increasing CO2 values, so that they can compare to the observed NEE and ecosystem respiration more directly, rather than comparing the (naturally offset/biased) equilibrium state of the model? I appreciate that the lack of a site and disturbance history would result in biases in the model simulations, but this spinup + historical simulation protocol is very common, and I presume is normally used to run CTEM for model inter-comparisons as well as climate change simulations? The authors state that their primary objective is to evaluate the temporal dynamics. But I do not see

any issue therefore with running a historical run – as is the often used protocol – and then stating more clearly that their only goal is to look at the temporal dynamics. In any case, the decision to only compare the model at its equilibrium state (as detailed in lines 421-425 for example) should be put in the methods, not in the results, so the reader is fully aware before they get to the results.

Section 3.2 Line 440: I am a bit confused as the stem's NSC pool does not get depleted in Figs 7-9c? It decreases a little, but not by a large amount as a fraction of its size? I also would expect that given the addition of NSC pools is the main focus of this study, the model should be evaluated at sites which do have NSC data.

Section 3.3 I find this section somewhat distracting given, aside from the last sentence, the differences between the original and modified model are not discussed much. In fact, the differences are very small. The authors note this, but do not provide any discussion as to why the change in seasonality of the simulated LAI does not alter energy fluxes more, as one might expect.

Discussion and conclusions

Aside from the conclusions part to this section, I find the rest of this section lacks a more in-depth discussion in places. There is some discussion of future perspectives to further improve the modeling of LAI (lines 513-525), and the possibility to include the other processes such as drought mortality and the N cycle due to the requirement to model N in leaf NSC pools (lines 552-554). However, there could also be more discussion of the results that might place them in a wider context. E.g. what are the implications for the wider modeling community? How do your results compare to ways NSC-related processes have been implemented in other NSC modeling studies (see review and references in Dietze et al., 2014). A discussion of any caveats to their work would also be useful. These might include some of the points I raised in my general comments, or a discussion about the uncertainty in NSC processes implemented and/or those that remain poorly understood (as the authors stated in the introduction).

References

Dietze, M.C., Sala, A., Carbone, M.S., Czimczik, C.I., Mantooth, J.A., Richardson, A.D. and Vargas, R., 2014. Nonstructural carbon in woody plants. Annual review of plant biology, 65, pp.667-687.

Melaas, E.K., Richardson, A.D., Friedl, M.A., Dragoni, D., Gough, C.M., Herbst, M., Montagnani, L. and Moors, E., 2013. Using FLUXNET data to improve models of springtime vegetation activity onset in forest ecosystems. Agricultural and Forest Meteorology, 171, pp.46-56.

Richardson, A.D., Anderson, R.S., Arain, M.A., Barr, A.G., Bohrer, G., Chen, G., Chen, J.M., Ciais, P., Davis, K.J., Desai, A.R. and Dietze, M.C., 2012. Terrestrial biosphere models need better representation of vegetation phenology: results from the North American Carbon Program Site Synthesis. Global Change Biology, 18(2), pp.566-584.

---

## Author Comment (AC2) · 31 Jul 2018

We appreciate reviewer #2's comments on our manuscript. Our point-by-point responses are included below. The reviewer's comments are indicated in an italic font and our responses are in a regular one.

 *Reviewer:*
*General Comments*

[Figure]

*The study by Asaadi et al. aims to improve the seasonal timing of LAI simulated by the CTEM model by including a representation of non-structural carbon (NSC) pools and fluxes in the model (in addition to a few other modifications). The new developments in the model are tested at three temperate broadleaved deciduous sites against LAI, carbon and energy flux observations. They show an improvement in the timing of various stages in the phenological cycle, and corresponding improvement in the timing of carbon fluxes (though limited improvement in energy fluxes). This is of use to the land surface modeling community as not all LSMs currently include specific NSC pools and related processes. As the authors have discussed, NSC processes are relevant for other components of biogeochemical cycling or ecosystem functioning, in addition to phenology that they focus on in this study (though this discussion could be expanded). The paper is clearly written and structured (although the goals of their model development work might be better stated as questions or hypotheses). However, I have some concerns about the lack of breadth of the study and depth of the analyses undertaken, which are detailed more in specific comments and outlined briefly here. 1) The addition of non-structural carbon pools and associated fluxes is the primary focus of the study; however, there are no observations of NSCs used to evaluate the authors' modifications to the model. I would have expected that any model modification would be tested against observations that are directly relevant to the new processes added in the model even though I appreciate that NSC data are scarce (see Dietze et al., 2014 for a review of previous such studies in the literature). Why was this not the case? Although the authors state the sites chosen were those with available LAI data, was it not possible to evaluate this model at any site that had observations of non-structural carbon pools (even if those data came from sites that did not also have LAI data)? The authors only chose three sites representing only one plant functional type. I would think there are a greater number of sites with LAI data that this model could be tested against.*

**Authors:**
As mentioned in many other studies as well as in Dietz et al. (2014), plants physiological processes are extraordinary complex and there is still a considerable debate and

lack of understanding regarding the behavior of NSC reserves. In addition, there are huge differences between plant species as to how they use their NSC reserves (e.g., Li et al., 2016; Wiley and Helliker, 2012; and Hoch et al., 2003). We are not aware of any long-term NSC measurements where meteorological data are available to drive the model with and LAI measurements are also available to evaluate the model. The relative size of the NSC pools (~5-10% of the total carbon pool in each component of a tree) is inferred from observations (e.g., Li et al., 2016) and the model aims to achieve this. The NSC-reallocation process during bud burst follows the classic source-sink paradigm. This study is the first attempt to include NSC pools and their associated processes in the CLASS-CTEM model and in our humble opinion model evaluation at three sites is enough to illustrate the proof of the concept. Evaluation of the model at regional and global scales will follow in due course but these are subjects of our future studies.

We have, however, followed on other recommendations of reviewer #2 as explained below in answers to his/her comments.

***Reviewer:***
*2) While the authors detail improvements in their modified model in comparison with observations (though less so for energy fluxes), it is not clear which of the model modifications made (detailed in Sections 2.1.1 to 2.1.4) are responsible for the improvements in the simulated carbon and energy fluxes. The authors could show the impact of each modification individually, before evaluating all together. I think the modeling community would appreciate knowing how each of the different modifications made to the model contributed to the overall improvement in the model. Such an analysis would help them ascertain if there are potential structural deficiencies in their own model, thus placing this work in a wider context.*

**Authors:**
Thank you for this suggestion. We have carried out such an analysis and this is illustrated for the Harvard Forest site in the figure below. A similar behavior was seen at

the other two sites. Panel (a) of the figure below indicates the resulting LAI after implementing the first step (i.e., reallocation of NSC during leaf out period, presented in section 2.1.2). The reallocation occurs only in early spring and as expected, it improves the ascending side of the simulated LAI during the same period. In panel (b), the effect of the second modification which has been discussed in section 2.1.3 is also included. As it can be seen, reducing the allocation fraction to leaves after summer solstice affects the LAI during the peak growing season and shifts the annual maximum LAI to earlier days of the year. Finally, the effect of the third modification step, section 2.1.4 in the text, is also included (panel (c) on the figure below). Increasing the lower air temperature threshold after summer solstice leads to an earlier leaf litter fall (not shown) and consequently an improvement on the descending side of the LAI curve. Together, the three modification steps lead on an overall improved CLASS-CTEM's simulated phenology.

*Reviewer:*
*3) The analysis lacks depth – namely, there is a lack of a rigorous quantitative evaluation of the modified model. It would be useful to include certain metrics to quantify the improvements simulated by the modified model (simple correlations for example). In addition, given the authors state that their primary goal is to address the issue of delayed leaf phenology, their analyses should be focused only on that question; general discussions of model behavior and magnitude of fluxes are distracting, especially given they have decided not to run a historical or transient simulation of the model after the spinup, with increasing $CO_2$ and climate.*

**Authors:**
We will add coefficient of correlation ($R^2$) and root-mean-squared error (RSME) to Figs. 4, 5, 6, and 10 when revising our manuscript. We, however, respectfully disagree that general discussion of carbon fluxes is distracting. While LAI is an important variable, it is the land-atmosphere $CO_2$, energy and water fluxes which determine the regional and global climate in an Earth system model. Assessment of carbon and energy fluxes is

an important part of any model evaluation exercise. We are unable to perform historical transient simulations because we do not have long-term meteorological data to drive the model with. The conclusions derived in the manuscript will not change even if we were able to perform transient historical simulations. Performing transient historical simulations would have provided the current observation-based annual positive values of net biome productivity to evaluate the model against but this is not the focus of our manuscript. Our past experience shows that steady state simulations allow an easier interpretation of model modifications. We will expand on this discussion when revising our manuscript.

*Reviewer:*
*4) The authors state in the discussion around lines 534-535 that the omission of NSC pools in the original model was a structural error. However, they do not definitively provide evidence to support their claim that the omission of NSC pools was a structural error. While their results show that this process can improve model LAI temporal dynamics, they have not conclusively shown that this is the only process that could be responsible for any discrepancies between the model and the observations, and therefore how important it is to add these specific processes. Incorporation of NSC pools and fluxes may not be the only process that can alleviate the problems in the simulated LAI. As they go on to state, biological systems are complex and difficult to represent with physical equations in models. To ensure that we do have the right model behavior, the processes we include must be rigorously tested against data corresponding to that process. Ideally, the authors would test alternative functions available in the literature for the processes they have implemented, in order to estimate the structural uncertainty associated with the new model developments. A Bayesian model selection framework could be used in order to select the most parsimonious model based on a model selection criterion (such as the Akaike Information Criterion – see Melaas et al., 2013 for an example). I would also be interested to see an analysis on the uncertainty related the parameters they have implemented. It might then be useful to discuss other NSC related processes that remain poorly understood that are not captured by their*

*new model.*

**Authors:**
An aspect of model testing that we omitted in the manuscript is that in the past we tried implementation of time-dependent maximum carboxylation capacity of Rubisco ($V_{cmax}$) as a function of the day length at each latitude. Since $V_{cmax}$ is typically much larger early on in the growing season than during the latter half of the season, we suspected this may fix the problem of delayed leaf out, but it did not. The fact that plants do indeed use their NSC pools for flushing new leaves and that such parameterizations did not help, indicates that omission of NSC pools in the original model version was indeed a structural error. Testing alternative available parameterizations is a big exercise which is beyond the scope of our study and parameterizations from other models require that they fit within the conceptual framework of our model.

*Reviewer:*
*5) The discussion lacks depth as to how the models they have implemented compare to other studies that have already implemented NSC models, as well as a discussion of any caveats to their modeling work related to the points I mention here. See specific comments.*

**Authors:**
Thank you for pointing this limitation which was also raised by the other reviewer. We will modify our manuscript to include references to and discussion of existing work on inclusion of NSC pools in ecosystem models and compare their conceptual model with ours.

*Reviewer:*
*Specific Comments*
*Introduction*
*Lines 109-111: Unless I have misunderstood, this model has been used in a phenology comparison at these sites (Richardson et al., 2012). If I have the right model, it*

*seems to me that the problems in the behavior of CTEM (for simulating LAI) shown in Richardson et al. are different to that in Anav. This shows that there might be other issues in the phenology models already implemented in CTEM due to differences between versions/parameterizations, without the addition of new processes/modifications to the model?*

**Authors:**

Results presented in Richardson et al. (2012) were produced using a very old version (which dates back to year 2005) of the model utilized by a university collaborator who coupled it with his own nitrogen module. The model has evolved considerably since then and includes several new parameterizations. As an example, the two simulated LAIs for the US-UMB site (Fig. 1 in Richardon et al. and Fig. 6 panel a in our study) are not similar.

*Reviewer:*
*Model, data and methods*
*Sections 2.1.1 and 2.1.2 There is a lack of references and/or reasoning for some of the mechanisms they are implementing in the models and the various assumptions they make in doing so (e.g. assumption that respiratory losses occur from non-structural part – line 176, and the references/reasoning behind formulation in equations 5, 6). In addition, the reasoning behind fixing certain parameter values needs to be detailed (e.g. why $\mu_i = 70$ line 191; beta in line 241). Were the parameter values found from the literature, or perhaps they were calibrated based on sensitivity studies or optimization experiments?*

**Authors:**

Thank you for pointing this out. We will add references and expand text to justify our parameterizations. For example, maintenance respiration which is temperature dependent occurs only from the nonstructural pools in agreement with observational and modelling studies (e.g., Hoch et al., 2003; Sperling et al., 2015; and Li et al., 2016) which show that plants' NSC stores become depleted due to excess respiration during

dormant season and during disturbances such as cold and drought stresses. Since biological processes have to be parameterized their parameters have to be calibrated to obtain realistic model behaviour. We will expand on this and discuss the basis for choosing the parameter values that we have used.

*Reviewer:*
*Line 186: maybe refer to Section 2.1.2 for how is Tj calculated?*

**Authors:**
Agreed.

*Reviewer:*
*Section 2.1.3 Are you referring to the fact that CLASS-CTEM has a flat peak of around 2 months in Anav et al., 2013 Fig 11, as opposed to the sharper (~1 month) peak seen in the observations and other models? In that case, it might be good to just state this in parentheses, as I was distracted by the fact that LAI simulated by CLASS-CTEM does not start to decline until long after the summer solstice (Anav et al., 2013 Fig 11) and much later than the observations. In any case, is there evidence in the literature that allocation fractions is modulated by day length – as represented in Section 2.1.3? No references are given to support the addition of this process. Could this slower rate of decline be due to incorrect parameters/processes related to senescence? Initially I was more distracted by the fact the seasonal cycle is delayed (out of phase) by a month or so. I am not therefore convinced if this correction factor based on day length presented in Section 2.1.3 is needed on top of other structural changes in the model.*

**Authors:**
We will make it explicitly clear the problems with simulated LAI in Anav et al. 2013. Yes, indeed there are two problems with simulated global mean LAI – it peaks later on with a much flatter peak, and it starts to increase later than observation-based LAI. We will also cite additional references in support of using summer solstice as a date to adjust our allocation fractions. Adjustments to allocation fraction for leaves after

summer solstice have also been made in other studies (e.g., see Eq. (6) of Gim et al., 2017). Summer solstice is a trigger for many other physiological processes. For instance, Luo et al. (2018) showed that summer solstice marks a seasonal shift in plant growth, leaf physiology, and foliage turnover in temperate and boreal forests. In earlier versions of the CLASS-CTEM model, continuous allocation of net primary productivity to leaves led to increase in LAI throughout the growing season rather than a constant or slowly decreasing LAI over the growing season. The effect of the adjustment which we have made to the allocation fraction to leaves after summer solstice is more obvious in the figure above which indicates the effect of each model parameterization separately.

***Reviewer:***
*Section 2.1.4 Similarly to Sections 2.1.1 (above), why is a value of 12 °C now used for $T_{cold}^{leaf}$? Is this based on the literature, or experiments, or a calibration exercise? Please give details and/or references as to how this value was chosen.*

**Authors:**
The value for $T_{cold}^{leaf}$ has been obtained based on a calibration exercise during our third modification step (section 2.1.4) and its individual effect will be shown in a supplementary figure based on the figures shown above. Model parameters depend on their parameterizations. In the CLASS-CTEM model, leaf loss due to cold stress begins when air temperature falls below thethreshold but its rate accelerates as temperature continues to drop. Leaf loss rate due to cold stress is maximum when temperature falls 5 °C below thethreshold.

***Reviewer:***
*Section 2.2.2 Please could you detail where you got the site meteorological data from, and which method and/or software you used to gap-fill the met data? Also, please could you detail why you chose to use a $CO_2$ concentration of 350 ppm (this is detailed around line 418 in the results, but needs to be put here). Finally, please could you detail how the LAI measurements were made at each site? Are there differences between sites? This information would be helpful for readers.*

**Authors:**
As mentioned on lines 113-115 of the discussion manuscript, gap-filled meteorological data are obtained from the Fluxnet network. We will move the part detailing the $CO_2$ concentration to section 2.2.2. LAI measurements are taken using an LAI-2000 plant canopy analyzer instrument (details are provided in Urbanski et al., 2007, Schmid et al., 2000, and Gough et al., 2008 for the Harvard Forest, Morgan Monroe, and U. of Michigan sites, respectively). We will make this clear when revising our manuscript.

**Reviewer:**
*Results*
*Figures 4-6: It would be good to state that both simulated and observed values represent averaged daily values across all years where data are present in the figure captions.*

**Authors:**
Agreed. We will mention this when revising figure captions.

**Reviewer:**
*Section 3.1 It would be helpful to have some metrics that show improvement (or lack thereof) between model versions for the full timeseries at each site. Even just RMSE or R would be helpful to quantify this and help put the results in context. This could be added to Table 3 for example.*

**Authors:**
We will add $R^2$ and RMSE to each figure when revising our manuscript.

**Reviewer:**
*Lines 382-384. It would be helpful if the authors showed a comparison of the observations and the model for each of the different modifications to the model that the authors have made in this study (as described in Sections 2.1.1-2.1.4), in addition to the overall improvement brought about by all modifications together. That way, other modeling groups can assess which modifications might be necessary for their own model – thus*

*making the study useful in a wider context. These may only be put supplementary figures or tables, but it would still be useful to discuss in the text.*

**Authors:**

As mentioned above and shown in the figure we have performed such an analysis and we will include this in the supplementary information.

*Reviewer:*

*Lines 390-391: This is true, and a good result. However, I also noted there seems to be an offset in the start of LAI and GPP in the observations at both US-MMS and US-UMB. At US-MMS the onset of LAI now better matches the observations, but there is now a bias towards a too early rise in GPP. Similarly, although the LAI at US-UMB now better matches the observations, and the GPP matches the observations very well, there is a still this offset. Why do you think that is? Is there a discrepancy between the two types of observations?*

**Authors:**

Indeed! The observations of GPP and LAI are not perfectly consistent with each other. The two sets of observations are made by different groups of people. This highlights the issue that when comparing model results with observations care must be taken to ensure that observations make sense. We will expand more on this aspect when revising our manuscript.

*Reviewer:*

*Lines 396-414: Why haven't the authors run a historical simulation after their spinup using increasing $CO_2$ values, so that they can compare to the observed NEE and ecosystem respiration more directly, rather than comparing the (naturally offset/biased) equilibrium state of the model? I appreciate that the lack of a site and disturbance history would result in biases in the model simulations, but this spinup + historical simulation protocol is very common, and I presume is normally used to run CTEM for model inter-comparisons as well as climate change simulations? The authors state*

*that their primary objective is to evaluate the temporal dynamics. But I do not see any issue therefore with running a historical run – as is the often used protocol – and then stating more clearly that their only goal is to look at the temporal dynamics. In any case, the decision to only compare the model at its equilibrium state (as detailed in lines 421-425 for example) should be put in the methods, not in the results, so the reader is fully aware before they get to the results.*

**Authors:**

Doing a transient historical simulation for each site requires $CO_2$ concentration data (which we have) as well as meteorological data (which we do not have). We mentioned this in section 2.2.2 but we will expand on this even more. Note that the increased ecosystem respiration at present day is not only due to increased GPP (associated with increasing $CO_2$) and therefore somewhat increased size of carbon pools (compared to what would be obtained from spinning a model at 350 ppm $CO_2$), but also the increased temperatures that ecosystems are subjected to over the historical period. Without meteorological data that shows a warming trend over the historical period, we would not be able to properly simulate increased respiration at present day. We perform transient historical simulations on a routine basis but chose not to do so for this study since our objective was to evaluate seasonal dynamics of LAI in a clean manner.

*Reviewer:*

*Section 3.2 Line 440: I am a bit confused as the stem's NSC pool does not get depleted in Figs 7-9c? It decreases a little, but not by a large amount as a fraction of its size? I also would expect that given the addition of NSC pools is the main focus of this study, the model should be evaluated at sites which do have NSC data.*

**Authors:**

Very little carbon is actually needed to construct leaves. In fact, on average, the total NSC pool for trees are estimated to be enough to completely rebuild the whole leaf canopy 1-4 times (Dietze et al., 2014). Even higher values have been mentioned in other studies (e.g., Hoch et al., 2003, and Mei et al., 2015).

*Reviewer:*

*Section 3.3 I find this section somewhat distracting given, aside from the last sentence, the differences between the original and modified model are not discussed much. In fact, the differences are very small. The authors note this, but do not provide any discussion as to why the change in seasonality of the simulated LAI does not alter energy fluxes more, as one might expect.*

**Authors:**

Actually, we discussed this briefly on lines 504-508 in the Discussion section but will expand on this and include it in section 3.3 as well. The reason for much lower effect of changes in LAI on latent heat fluxes (or equivalently evapotranspiration), than for GPP, is that while GPP is solely determined by LAI, evaporation also occurs from soil and from intercepted water on canopy leaves. If evaporative demand for a given available energy cannot be met by transpiration then the demand is met by evaporation from soil. Although, of course, soil evaporation also depends on soil moisture in the top soil layer. But generally speaking, the different components of evapotranspirative flux are able to compensate for each other. As a result, small changes in LAI do not affect latent heat flux as much as they do GPP.

*Reviewer:*

*Discussion and conclusions*

*Aside from the conclusions part to this section, I find the rest of this section lacks a more in-depth discussion in places. There is some discussion of future perspectives to further improve the modeling of LAI (lines 513-525), and the possibility to include the other processes such as drought mortality and the N cycle due to the requirement to model N in leaf NSC pools (lines 552-554). However, there could also be more discussion of the results that might place them in a wider context. E.g. what are the implications for the wider modeling community? How do your results compare to ways NSC-related processes have been implemented in other NSC modeling studies (see review and references in Dietze et al., 2014). A discussion of any caveats to their*

*work would also be useful. These might include some of the points I raised in my general comments, or a discussion about the uncertainty in NSC processes implemented and/or those that remain poorly understood (as the authors stated in the introduction).*

**Authors:**
We will add more discussion along the lines suggested by the reviewer including comparison to other models who have implemented NSC pools. We will also include discussion of implications for the wider modeling community when revising our manuscript including in particular the mention of our failed approaches taken to fix the problem of delayed phenology.

**References:**

Anav, A., and Coauthors, 2013: Evaluating the land and ocean components of the global carbon cycle in the CMIP5 earth system models. *J. Climate,* **26,** 6801–6843.

Dietze, M. C., A. Sala, M. S. Carbone, C. I. Czimczik, J. A. Mantooth, A. D. Richardson, and R. Vargas, 2014: Nonstructural carbon in woody plants. *Annual review of plant biology,* **65,** 667-687.

Gim, H. J., S. K. Park, M. Kang, B. M. Thakuri, J. Kim, and C. H. Ho, 2017: An improved parameterization of the allocation of assimilated carbon to plant parts in vegetation dynamics for Noah−MP. *Journal of Advances in Modeling Earth Systems,* **9(4),** 1776–1794.

Gough, C. M., C. S.Vogel, H. P. Schmid, H. B. Su, P. S. Curtis, 2008: Multi-year convergence ofbiometric and meteorological estimates of forest carbon storage.

*Agricultural and Forest Meteorology,* **148,** 158–170.

Hoch, G., A. Richter, C. Körner, 2003: Non-structural carbon compounds in temperate forest trees. *Plant Cell Environ.,* **26,** 1067–1081.

Li, N., N. He, G. Yu, Q. Wang, J. Sun, 2016: Leaf non-structural carbohydrates regulated by plant functional groups and climate: evidences from a tropical to cold-temperate forest transect. *Ecological Indicators,* **62,** 22–31.

Luo, T., X. Liu, L. Zhang, X. Li, Y. Pan, and I. J. Wright, 2018: Summer solstice marks a seasonal shift in temperature sensitivity of stem growth and nitrogen-use efficiency in cold-limited forests. *Agricultural and Forest Meteorology,* **248,** 469–478.

Mei, L., Y. Xiong, J. Gu, Z. Wang, D. Guo, 2015: Whole-tree dynamics of non-structural carbohydrate and nitrogen pools across different seasons and in response to girdling in two temperate trees. *Oecologia,* **177,** 333–344.

Richardson, A. D., R. S. Anderson, M. A. Arain, A. G. Barr, G. Bohrer, G. Chen, J. M. Chen, P. Ciais, K. J. Davis, A. R. Desai, and M. C. Dietze, 2012: Terrestrial biosphere models need better representation of vegetation phenology: results from the North American Carbon Program Site Synthesis. *Global Change Biology,* **18(2),** 566-584.

Schmid, H. P., C. S. B. Grimmond, F. Cropley, B. Offerle, H. B. Su, 2000: Measurements of $CO_2$ and energy fluxes over a mixed hardwood forest in the mid-western United States. *Agricultural and Forest Meteorology,* **103,** 357–374.

Sperling, O., J. M. Earles, F. Secchi, J. Godfrey, M. A. Zwieniecki, 2015: Frost induces-respiration and accelerates carbon depletion in trees. *PLoS One,* **10**:e0144124.

Urbanski, S., and Co-authors: Factors controlling CO2 exchange on timescales from hourly to decadal at Harvard Forest. *Journal of Geophysical Research-Biogeosciences,* **112,** G02020.

Wiley, E., and B. Helliker, 2012: A re-evaluation of carbon storage in trees lends greater support for carbon limitation to growth. *New Phytologist Trus,* **195,** 285-289.

**BGD**
[Figure]

**Leaf Area Index**

a

- CLASS-CTEM Original (annual mean= 2.05)
- CLASS-CTEM Modified (annual mean= 2.47)
- •••  Fluxnet (annual mean= 1.85)

LAI ($m^2/m^2$)

Day of year

**Leaf Area Index**

b

- CLASS-CTEM Original (annual mean= 2.05)
- CLASS-CTEM Modified (annual mean= 2.28)
- •••  Fluxnet (annual mean= 1.85)

LAI ($m^2/m^2$)

Day of year

**Leaf Area Index**

c

- CLASS-CTEM Original (annual mean= 2.05)
- CLASS-CTEM Modified (annual mean= 1.92)
- •••  Fluxnet (annual mean= 1.85)

LAI ($m^2/m^2$)

Day of year

**Fig. 1.**

[Figure]

[Figure]

---

## Author Response (AR1)

We appreciate the time and constructive comments from anonymous reviewers and the editor on our manuscript. A point-by-point summary of the modifications that we have declared to make before in response to the reviewers' comments are included below. The reviewers' comments are indicated in italic font and our responses are in regular font.

***Reviewer #1***
* * *
*Reviewer:*

*These are my comments of the work entitled: An improved parameterization of leaf area index (LAI) seasonality in the Canadian Land Surface Scheme (CLASS) and Canadian Terrestrial Ecosystem Model (CTEM) modelling framework. Firstly, I want to thank the authors for the work done. I felt the manuscript quite interesting. This paper describes the addition of the Non Structural Carbohydrates (NSC) module, to the CLASS-CTEM model. NSC module allows to better represent Leaf Area seasonality, as well as to provide a mobile carbohydrate pool to the trees to increase its resilience to disturbances in absence of photosynthesis. It is tested in three Fluxnet sites, where GPP, LAI, and heat fluxes (Incident radiation, latent heat and sensible heat) model projections are contrasted against real data. In my opinion, this is an interesting, thoroughfull work, where the authors clearly demonstrate that the addition of the NSC module clearly improves model performance. My major concerns about the present paper are about its novelty. Currently most of process-based forest simulation models does include the NSC module (Fontes et al., 2010), in a similar way than the new module for the CLASS-CTEM model. So, in my opinion, your current manuscript doesn't clarify the novelty of your work. Furthermore, throughout your manuscript there is little reference to other models that include this key compartment, and I think it would be a nice element to include in the discussion, as there is plenty of other works in which the addition of NSC in a given model clearly improves its performance.*

**Authors:**

References to existing work on inclusion of NSC pools in ecosystem models have been added to the manuscript (lines 102-121). Although NSC pools have been included in other process-based models, the CLASS-CTEM model lacked this part and the present study is the first effort to address this problem. While we do not claim novelty for the present study, we have clarified and stressed the importance of NSC pools even more in the revised version of our manuscript (lines 222-227, 453-456, 571-577, 608-612, 619-629).

*Reviewer:*

*Specific considerations:*
*I felt a little lacking how the Maintenance respiration was calculated in CLASS-CTEM. I've seen other reviewers asking for the same point, and I feel like an addition of the maintenance respiration formula as well as the assumptions of the model about this process would improve significantly the paper. Besides, I have a couple of questions about maintenance respiration: it is dependent on temperature? It is assumed the same respiration rate for the structural and non-structural carbohydrates?*
*In lines 175-177, you state that respiratory carbon loses are assumed to occur from the non-structural*

*part. Does it mean that structural carbon is not accounted in maintenance respiration? I guess you did not mean that, but as it is stated, it may lead to misinterpretations.*

**Authors:**

The text was modified and some clarifications were added in response to the editor's review (lines 178-185 and 216-218) in the context of maintenance respiration. The model's maintenance respiration along with other processes have already been discussed in previous publications (e.g., section A3.1 in Melton and Arora, 2016).

Yes, indeed maintenance respiration is temperature dependent. In the modified (i.e., NSC-added) version of the model presented in this study, maintenance respiration occurs only from the non-structural pools consistent with observational and modelling studies (e.g., Hoch et al., 2003; Sperling et al., 2015; and Li et al., 2016) which show that plants' NSC stores become depleted during dormant season and cold and drought stresses due to excess respiration (lines 196-199).

*Reviewer:*

*It is a minor issue, but, in general, I think your explanations about Leaf Area (LA) importance upon photosynthesis. However, I think you are wrong when referring to them as LAI (for example, lines 1, 63). LAI doesn't perform photosynthesis, it is the Leaf Area, that does it. LAI is just an explanatory index about the surface of leaf area per unit of surface.*

**Authors:**

We feel this is a terminology issue. While it is true that photosynthesis is a function of the total "leaf area", several of the model's parameterizations use leaf area index since all model processes are simulated per unit area. For example, CLASS-CTEM model uses the Beer-Lambert law ($1 - e^{-k(LAI)}$, where k is a vegetation-dependent light extinction coefficient) to scale photosynthesis from the leaf to the canopy level. The related text has been modified (lines 176-177) to reflect this.

*Reviewer:*

*In point 2.1.1 (Reallocation of non-structural carbon during leaf out period), you do state that "after reaching a threshold LAI, NPP is allocated to stem and roots in addition to leaves". I could not find in your work how these compartments are developed. Do your model follow any predefined rule (e.g. the Pipe model rule, Shinozaky, 1974)? Or they are equally allocated throughout the tree compartments according to a predefined rate?*

*Line 451. Again, following which rule, besides the "after reaching a LAI threshold", are the carbohydrates allocated through the three compartments? A fixed rate? A mechanistic rule?*

**Authors:**

While the model's several processes have been described before in Melton and Arora (2016), additional details are added at lines 276-285 to clarify these allocation-related questions raised by the reviewer. In brief, the model uses dynamically calculated allocation fractions for leaves, stems and roots based on light, water and leaf phenological status of vegetation. The preferential allocation of carbon to the different tissue pools is based on three assumptions: (i) if soil moisture is limiting, carbon should be preferentially allocated to roots for greater access to water, (ii) if LAI is low, carbon should be allocated to leaves for enhanced photosynthesis and finally (iii) carbon is allocated to the stem to increase vegetation height and lateral spread of vegetation when the increase in LAI results in a decrease in light penetration. The dynamically calculated allocation fractions are further modified to ensure that root to shoot ratio is realistic and that there is enough woody biomass to support green biomass.

*Reviewer:*

*Line 372-373. I would remove the sentence "The figure legends, in addition to identifying the two mode versions and observations, also show the mean annual value of the quantity plotted", and I would include it in the figure footnote.*

**Authors:**

Agreed.

*Reviewer:*

*Lines 419-420. Your sentence assumes an equilibrium between the atmospheric $CO_2$ and the biosphere. Maybe is far beyond the discussion of your paper, but I think that this is not strictly true, as there has been previous works identifying the instability of the atmosphere-biosphere complex (e.g. Higgins et al., 2002). It is a minor change, but I would suggest to erase the "currently" in the sentence, thus indicating the responsive nature of biosphere to historical changes in atmospheric $CO_2$.*

**Authors:**

Agreed.

*Reviewer:*

*Point 3.3. Here, you find that you do overestimate latent heat when modelling your three forests. Are there any research papers about evapotranspiration experiments in those forests? If they are, maybe you should transform latent heat into evapotranspiration values, so you can compare them to your data, and you might then have a better explanation about why does your model overestimates so high the evapotranspiration (Latent heat). In addition, how do this relate to your overestimation of Leaf Area? I see a little discussion about it in lines 508-512, but I think you minimize the effect that you overestimate the LAI during the growing season by about 2 $m^2/m^2$ in each plot, and this affect both the evapotranspiration at canopy level, but also to the evaporative energy available at ground level.*

**Authors:**

As the reviewer noted evapotranspiration and latent heat fluxes are the same quantity but in different units. Evapotranspiration is expressed in water units (mm) and latent heat flux is expressed in energy units (W/m$^2$). At the flux net sites latent heat flux is available but as mentioned in section 3.3 of the manuscript it suffers from energy balance closure. So it is difficult to conclusively determine how much of this overestimation is due to overestimation by the model and how much of it is due to bias in the observed data. Our experience with the CLASS-CTEM model in the past has been that in locations where soil moisture constraint is not very large, as is the case at these three sites, total evapotranspiration is controlled by available energy. Hence the expected seasonality in latent heat flux is characterized by higher values during summer and lower during winter at these sites. Since the latent heat flux at these three sites is primarily controlled by available energy the resulting implication is that if evaporative demand cannot be met by transpiration then it will be met by evaporation from the soil. As a result, changes in LAI do not significantly affect total evapotranspiration (or latent heat flux) but change the partitioning of evapotranspiration flux coming from transpiration, evaporation of intercepted water on canopy leaves, and evaporation from the soil. Therefore, had the model simulated lower LAI than it currently does, then the latent heat flux would not have been significantly different from its current values. Additional discussion to make this point has been added (lines 533-543, 587-599).

*Reviewer:*

*Lines 519 to 523. Please, revise the Vmax concept: as states the original paper from Kattge et al, (2009), Vmax is the maximum carboxilation capacity, not the maximum photosynthetic rate. In addition: you are justifying a tautology: This is the parameter that we want to apply to Vmax, so we adjust all other model parameters to fit the results according to this Vmax value. In addition, you finish the sentence with "it is possible that the average Vmax value derived by Kattge et al. (2009) is not representative of [...]". I agree that mean Vmax value not representing correctly your forest performance is a possible explanation, but I would rather discuss that Vmax is not the only constrain to photosynthesis, as Jmax is also limiting assimilation rate.*

**Authors:**

Yes, indeed Vmax is the maximum carboxilation capacity. Thank you for catching this. The text has been modified (line 585). The CLASS-CTEM model uses the Farquhar approach for modelling photosynthesis, and therefore photosynthesis is limited by carboxilation capacity but also light and transport capacity limited rates. However, Vmax remains a very strong parameter that controls photosynthesis in the model. We have clarified this aspect in revising our manuscript (lines 584-599, 608-612).

We did not adjust other model parameters to accommodate Kattge et al. (2009) Vmax values. In fact, the very first version of the model used a value of Vmax of 65 umol $CO_2$/m$^2$s for its broadleaf cold deciduous PFT which is close to the value of 57 umol $CO_2$/m$^2$s suggested by Kattge et al. (2009).

It is probably worth mentioning that at the global scale the performance of CLASS-CTEM model when implemented in the Canadian Earth System Model (CanESM2) is fairly realistic. Figure 11 of Anav et al. (2013), who compared different Earth system models used in the phase 5 of the coupled model intercomparison project (CMIP5), shows that the CLASS-CTEM's simulated "annual global mean" LAI is among one of the best estimates compared to other models. Figure 9 of Anav et al. (2013) shows that the model also does a good job in simulating global annual GPP.

*Reviewer:*

*Figures 4-6. I would expand a little the footnote, to include the information that results are represented as averaged daily values. In addition, I would consider to change the Leaf Area Index inner pannel to represent the median values during the vegetative period rather than average values, as I think they would be more indicative of the similarities-differences between Fluxnet measurements and model outputs.*

**Authors:**

Thank you for pointing this out. The figures' footnotes are modified.

*Reviewer #2*

*Reviewer:*

General Comments

*The study by Asaadi et al. aims to improve the seasonal timing of LAI simulated by the CTEM model by including a representation of non-structural carbon (NSC) pools and fluxes in the model (in addition to a few other modifications). The new developments in the model are tested at three temperate broadleaved deciduous sites against LAI, carbon and energy flux observations. They show an improvement in the timing of various stages in the phenological cycle, and corresponding improvement in the timing of carbon fluxes (though limited improvement in energy fluxes).*

*This is of use to the land surface modeling community as not all LSMs currently include specific NSC pools and related processes. As the authors have discussed, NSC processes are relevant for other components of biogeochemical cycling or ecosystem functioning, in addition to phenology that they focus on in this study (though this discussion could be expanded).*

*The paper is clearly written and structured (although the goals of their model development work might be better stated as questions or hypotheses). However, I have some concerns about the lack of breadth of the study and depth of the analyses undertaken, which are detailed more in specific comments and outlined briefly here.*

*1) The addition of non-structural carbon pools and associated fluxes is the primary focus of the study; however, there are no observations of NSCs used to evaluate the authors' modifications to the model. I would have expected that any model modification would be tested against observations that are directly relevant to the new processes added in the model even though I appreciate that NSC data are scarce (see Dietze et al., 2014 for a review of previous such studies in the literature). Why was this not the case? Although the authors state the sites chosen were those with available LAI data, was it not possible to evaluate this model at any site that had observations of non-structural carbon pools (even if those data came from sites that did not also have LAI data)? The authors only chose three sites representing only one plant functional type. I would think there are a greater number of sites with LAI data that this model could be tested against.*

**Authors:**

As mentioned in many other studies as well as in Dietz et al. (2014), plants physiological processes are extraordinary complex and there is still a considerable debate and lack of understanding regarding the behavior of NSC reserves. In addition, there are huge differences between plant species as to how they use their NSC reserves (e.g., Li et al., 2016; Wiley and Helliker, 2012; and Hoch et al., 2003). We are not aware of any long-term NSC measurements where meteorological data are available to drive the model with and LAI measurements are also available to evaluate the model. The relative size of the NSC pools (~5-10% of the total carbon pool in each component of a tree) is inferred from observations (e.g., Li et al., 2016) and the model aims to achieve this. The NSC-reallocation process during bud burst follows the classic source-sink paradigm. This study is the first attempt to include NSC pools and their associated processes in the CLASS-CTEM model and in our humble opinion model evaluation at three sites is enough to illustrate the proof of the concept. Evaluation of the model at regional and global scales will follow in due course but these are subjects of our future studies.

We have, however, followed on other recommendations of reviewer #2 as explained below in answers to his/her comments.

*Reviewer:*
*2) While the authors detail improvements in their modified model in comparison with observations (though less so for energy fluxes), it is not clear which of the model modifications made (detailed in Sections 2.1.1 to 2.1.4) are responsible for the improvements in the simulated carbon and energy fluxes. The authors could show the impact of each modification individually, before evaluating all together. I think the modeling community would appreciate knowing how each of the different modifications made to the model contributed to the overall improvement in the model. Such an analysis would help them ascertain if there are potential structural deficiencies in their own model, thus placing this work in a wider context.*

**Authors:**

Thank you for this suggestion. We have carried out such an analysis and this is illustrated for the Harvard Forest site in the supplementary information. A similar behavior was seen at the other two sites.

*Reviewer:*
*3) The analysis lacks depth – namely, there is a lack of a rigorous quantitative evaluation of the modified model. It would be useful to include certain metrics to quantify the improvements simulated by the modified model (simple correlations for example). In addition, given the authors state that their primary goal is to address the issue of delayed leaf phenology, their analyses should be focused only on that question; general discussions of model behavior and magnitude of fluxes are distracting, especially given they have decided not to run a historical or transient simulation of the model after the*

*spinup, with increasing $CO_2$ and climate.*

**Authors:**

The coefficient of determination ($R^2$) and root-mean-squared error (RMSE) have been added to Figs. 4, 5, 6, and 10 in the revised version of the manuscript. We, however, respectfully disagree that general discussion of carbon fluxes is distracting. Lines 596-599 justify the inclusion of carbon fluxes.

The rationale for not performing historical transient simulations is discussed in lines 390-400 and 565-570. The conclusions derived in the manuscript will not change even if we were able to perform transient historical simulations. Performing transient historical simulations would have provided the current observation-based annual positive values of net biome productivity to evaluate the model against but this is not the focus of our manuscript.

*Reviewer:*

*4) The authors state in the discussion around lines 534-535 that the omission of NSC pools in the original model was a structural error. However, they do not definitively provide evidence to support their claim that the omission of NSC pools was a structural error. While their results show that this process can improve model LAI temporal dynamics, they have not conclusively shown that this is the only process that could be responsible for any discrepancies between the model and the observations, and therefore how important it is to add these specific processes. Incorporation of NSC pools and fluxes may not be the only process that can alleviate the problems in the simulated LAI. As they go on to state, biological systems are complex and difficult to represent with physical equations in models. To ensure that we do have the right model behavior, the processes we include must be rigorously tested against data corresponding to that process. Ideally, the authors would test alternative functions available in the literature for the processes they have implemented, in order to estimate the structural uncertainty associated with the new model developments. A Bayesian model selection framework could be used in order to select the most parsimonious model based on a model selection criterion (such as the Akaike Information Criterion – see Melaas et al., 2013 for an example). I would also be interested to see an analysis on the uncertainty related the parameters they have implemented. It might then be useful to discuss other NSC related processes that remain poorly understood that are not captured by their new model.*

**Authors:**

An aspect of model testing that we omitted in the manuscript is that in the past we tried implementation of time-dependent maximum carboxylation capacity of Rubisco ($V_{c,max}$) as a function of the day length at each latitude. Since $V_{c,max}$ is typically much larger early on in the growing season than during the latter half of the season, we suspected this may fix the problem of delayed leaf out, but it did not. The fact that plants do indeed use their NSC pools for flushing new leaves and that such parameterizations did not help, indicates that omission of NSC pools in the original model version was indeed a structural error. Testing alternative available parameterizations is a big exercise which is beyond the scope of our study and parameterizations from other models require that they fit within the conceptual framework of our model.

*Reviewer:*
*5) The discussion lacks depth as to how the models they have implemented compare to other studies that have already implemented NSC models, as well as a discussion of any caveats to their modeling work related to the points I mention here. See specific comments.*

**Authors:**
Thank you for pointing this limitation which was also raised by the other reviewer. We have modified our manuscript to include references to existing works which have included NSC pools in ecosystem models (lines 102-121).

*Reviewer:*
*Specific Comments*
*Introduction*
*Lines 109-111: Unless I have misunderstood, this model has been used in a phenology comparison at these sites (Richardson et al., 2012). If I have the right model, it seems to me that the problems in the behavior of CTEM (for simulating LAI) shown in Richardson et al. are different to that in Anav. This shows that there might be other issues in the phenology models already implemented in CTEM due to differences between versions/parameterizations, without the addition of new processes/modifications to the model?*

**Authors:**
Results presented in Richardson et al. (2012) were produced using a very old version (which dates back to year 2005) of the model utilized by a university collaborator who coupled it with his own nitrogen module. The model has evolved considerably since then and includes several new parameterizations. As an example, the two simulated LAIs for the US-UMB site (Fig. 1 in Richardon et al. and Fig. 6 panel a in our study) are not similar.

*Reviewer:*
*Model, data and methods*
*Sections 2.1.1 and 2.1.2 There is a lack of references and/or reasoning for some of the mechanisms they are implementing in the models and the various assumptions they make in doing so (e.g. assumption that respiratory losses occur from non-structural part – line 176, and the references/reasoning behind formulation in equations 5, 6).*
*In addition, the reasoning behind fixing certain parameter values needs to be detailed (e.g. why $\mu_i = 70$ line 191; beta in line 241). Were the parameter values found from the literature, or perhaps they were calibrated based on sensitivity studies or optimization experiments?*

**Authors:**

Thank you for pointing this out. The related text has been modified and more references and justifications have been added (lines 196-197, 213-221, 230-232, 306-310). For example, maintenance respiration which is temperature dependent occurs only from the nonstructural pools in agreement with observational and modelling studies (e.g., Hoch et al., 2003; Sperling et al., 2015; and Li et al., 2016) which show that plants' NSC stores become depleted due to excess respiration during dormant season and during disturbances such as cold and drought stresses. Since biological processes have to be parameterized their parameters have to be calibrated to obtain realistic model behaviour.

*Reviewer:*

*Line 186: maybe refer to Section 2.1.2 for how is Tj calculated?*

**Authors:**

Agreed.

*Reviewer:*

*Section 2.1.3 Are you referring to the fact that CLASS-CTEM has a flat peak of around 2 months in Anav et al., 2013 Fig 11, as opposed to the sharper (~1 month) peak seen in the observations and other models? In that case, it might be good to just state this in parentheses, as I was distracted by the fact that LAI simulated by CLASS-CTEM does not start to decline until long after the summer solstice (Anav et al., 2013 Fig 11) and much later than the observations. In any case, is there evidence in the literature that allocation fractions is modulated by day length – as represented in Section 2.1.3? No references are given to support the addition of this process. Could this slower rate of decline be due to incorrect parameters/processes related to senescence? Initially I was more distracted by the fact the seasonal cycle is delayed (out of phase) by a month or so. I am not therefore convinced if this correction factor based on day length presented in Section 2.1.3 is needed on top of other structural changes in the model.*

**Authors:**

Yes, indeed there are two problems with simulated global mean LAI – it peaks later on with a much flatter peak, and it starts to increase later than observation-based LAI.
We have modified the related text to clarify this (lines 288-291). Also, additional references have been cited in support of using summer solstice as a date to adjust our allocation fractions (lines 306-310).
In earlier versions of the CLASS-CTEM model, continuous allocation of net primary productivity to leaves led to increase in LAI throughout the growing season rather than a constant or slowly decreasing LAI over the growing season. The effect of the adjustment which we have made to the allocation fraction to leaves after summer solstice is more obvious in the supplementary information which highlights the effect of each model parameterization when implemented incrementally.

*Reviewer:*

*Section 2.1.4 Similarly to Sections 2.1.1 (above), why is a value of 12°C now used for Tleaf_cold? Is this based on the literature, or experiments, or a calibration exercise? Please give details and/or references as to how this value was chosen.*

**Authors:**

The value for $T_{cold}^{leaf}$ has been obtained based on a calibration exercise during our third modification step (section 2.1.4) and its individual effect is shown in supplementary information provided. Model parameters depend on their parameterizations. In the CLASS-CTEM model, leaf loss due to cold stress begins when air temperature falls below the $T_{cold}^{leaf}$ threshold but its rate accelerates as temperature continues to drop. Leaf loss rate due to cold stress is maximum when temperature falls 5 °C below the $T_{cold}^{leaf}$ threshold.

*Reviewer:*

*Section 2.2.2 Please could you detail where you got the site meteorological data from, and which method and/or software you used to gap-fill the met data? Also, please could you detail why you chose to use a $CO_2$ concentration of 350 ppm (this is detailed around line 418 in the results, but needs to be put here). Finally, please could you detail how the LAI measurements were made at each site? Are there differences between sites? This information would be helpful for readers.*

**Authors:**

As mentioned on lines 384-386 of the discussion manuscript, gap-filled meteorological data are obtained from the Fluxnet network. The part detailing the $CO_2$ concentration has been moved to section 2.2.2 (lines 390-400). LAI measurements are taken using an LAI-2000 plant canopy analyzer instrument (lines 347-350).

*Reviewer:*

*Results*
*Figures 4-6: It would be good to state that both simulated and observed values represent averaged daily values across all years where data are present in the figure captions.*

**Authors:**

Agreed. The figure captions have been modified.

*Reviewer:*

*Section 3.1 It would be helpful to have some metrics that show improvement (or lack thereof) between model versions for the full timeseries at each site. Even just RMSE or R would be helpful to quantify this and help put the results in context. This could be added to Table 3 for example.*

**Authors:**

The coefficient of determination ($R^2$) and root-mean-squared error (RMSE) have been added to Figs. 4, 5, 6, and 10.

*Reviewer:*

*Lines 382-384. It would be helpful if the authors showed a comparison of the observations and the model for each of the different modifications to the model that the authors have made in this study (as described in Sections 2.1.1-2.1.4), in addition to the overall improvement brought about by all modifications together. That way, other modeling groups can assess which modifications might be necessary for their own model – thus making the study useful in a wider context. These may only be put supplementary figures or tables, but it would still be useful to discuss in the text.*

**Authors:**

As mentioned above, we have performed such an analysis and included it in the supplementary information.

*Reviewer:*

*Lines 390-391: This is true, and a good result. However, I also noted there seems to be an offset in the start of LAI and GPP in the observations at both US-MMS and US-UMB. At US-MMS the onset of LAI now better matches the observations, but there is now a bias towards a too early rise in GPP. Similarly, although the LAI at US-UMB now better matches the observations, and the GPP matches the observations very well, there is a still this offset. Why do you think that is? Is there a discrepancy between the two types of observations?*

**Authors:**

Indeed! The observations of GPP and LAI are not perfectly consistent with each other. The two sets of observations are made by different groups of people. This highlights the issue that when comparing model results with observations care must be taken to ensure that observations make sense. This aspect is now discussed in lines 442-452.

*Reviewer:*

*Lines 396-414: Why haven't the authors run a historical simulation after their spinup using increasing $CO_2$ values, so that they can compare to the observed NEE and ecosystem respiration more directly, rather than comparing the (naturally offset/biased) equilibrium state of the model? I appreciate that the lack of a site and disturbance history would result in biases in the model simulations, but this spinup + historical simulation protocol is very common, and I presume is normally used to run CTEM for model inter-comparisons as well as climate change simulations? The authors state that their primary objective is to evaluate the temporal dynamics. But I do not see any issue therefore with running a historical run – as is the often used protocol – and then stating more clearly that their only goal is to look at the temporal dynamics. In any case, the decision to only compare the model at its equilibrium state (as detailed in lines 421-425 for example) should be put in the methods, not in the*

*results, so the reader is fully aware before they get to the results.*

**Authors:**

Doing a transient historical simulation for each site requires $CO_2$ concentration data (which we have) as well as meteorological data (which we do not have). While we mentioned this in our original manuscript we have now expanded on it even more (lines 390-400 and 565-570) . Note that the increased ecosystem respiration at present day is not only due to increased GPP (associated with increasing $CO_2$) and therefore somewhat increased size of carbon pools (compared to what would be obtained from spinning a model at 350 ppm $CO_2$), but also the increased temperatures that ecosystems are subjected to over the historical period. Without meteorological data that shows a warming trend over the historical period, we would not be able to properly simulate increased respiration at present day. We perform transient historical simulations on a routine basis but chose not to do so for this study since our objective was to evaluate seasonal dynamics of LAI in a clean manner.

*Reviewer:*

*Section 3.2 Line 440: I am a bit confused as the stem's NSC pool does not get depleted in Figs 7-9c? It decreases a little, but not by a large amount as a fraction of its size? I also would expect that given the addition of NSC pools is the main focus of this study, the model should be evaluated at sites which do have NSC data.*

**Authors:**

Very little carbon is actually needed to construct leaves. In fact, on average, the total NSC pool for trees are estimated to be enough to completely rebuild the whole leaf canopy 1-4 times (Dietze et al., 2014). Even higher values have been mentioned in other studies (e.g., Hoch et al., 2003, and Mei et al., 2015).

*Reviewer:*

*Section 3.3 I find this section somewhat distracting given, aside from the last sentence, the differences between the original and modified model are not discussed much. In fact, the differences are very small. The authors note this, but do not provide any discussion as to why the change in seasonality of the simulated LAI does not alter energy fluxes more, as one might expect.*

**Authors:**

This point has been discussed briefly before in the Discussion section but we have added additional text in section 3.3 as well (lines 533-543). The reason for much lower effect of changes in LAI on latent heat fluxes (or equivalently evapotranspiration), than for GPP, is that while GPP is solely determined by LAI, evaporation also occurs from soil and from intercepted water on canopy leaves. If evaporative demand for a given available energy cannot be met by transpiration then the demand is met by evaporation from soil. Although, of course, soil evaporation also depends on soil moisture in the top soil layer. But generally speaking, the different components of evapotranspirative flux are able to compensate for each other. As a result, small changes in LAI do not affect latent heat flux as much as they do GPP.

*Reviewer:*

*Discussion and conclusions*

*Aside from the conclusions part to this section, I find the rest of this section lacks a more in-depth discussion in places. There is some discussion of future perspectives to further improve the modeling of LAI (lines 513-525), and the possibility to include the other processes such as drought mortality and the N cycle due to the requirement to model N in leaf NSC pools (lines 552-554). However, there could also be more discussion of the results that might place them in a wider context. E.g. what are the implications for the wider modeling community? How do your results compare to ways NSC-related processes have been implemented in other NSC modeling studies (see review and references in Dietze et al., 2014). A discussion of any caveats to their work would also be useful. These might include some of the points I raised in my general comments, or a discussion about the uncertainty in NSC processes implemented and/or those that remain poorly understood (as the authors stated in the introduction).*

**Authors:**

More discussion along the lines suggested by the reviewer, including comparison to other models which have implemented NSC pools, has been added to the text (lines 102-121). We have also mentioned some of our failed approaches taken to fix the problem of delayed phenology (lines 608-612).

$$\Gamma = \left[ \frac{d}{d + (d_{max} - d)\, 0.5 \left( \tanh\left( \frac{\pi}{180}(20\phi - 800) \right) + 1 \right)} \right]^{20} \tag{8}$$

$$d = 24 - \frac{24}{\pi} acos\left[ max\left( -1, min\left( \frac{sin\phi \sin\delta_c}{cos\phi cos\delta_c}, 1 \right) \right) \right] \tag{9}$$

where $d$ is the day length at latitude $\phi$ (radian), $d_{max}$ is its maximum value (hour), and $\delta_c$ (radians) is solar declination. $\Gamma$ varies between 0 and 1 and its behaviour in Figure 2 shows how allocation to leaves is reduced at a faster (slower) rate closer to poles (equator) after summer solstice in the northern hemisphere (June 21). Below 30°N in the northern hemisphere equation (8) yields $\Gamma = 1$ so allocation fraction for leaves is not modified. Deciduousness due to day length and temperature typically does not occur in tropics where it is primarily controlled by soil moisture. Neither do broadleaf deciduous cold trees typically exist in the tropics. Similar behaviour is obtained for the southern hemisphere after

December 21. Since the allocation fractions for leaves, stem, and root components should add to 1, a decrease in allocation fraction for leaves implies an increase in allocation fraction for stem and root components in the modified version of the model. The use of summer solstice to initiate changes in plant behavior is reasonable since summer solstice is a trigger for many plant physiological processes.

Adjustments to allocation fraction for leaves after summer solstice have also been made by Gim et al.

(2017) (their equation (6)). Luo et al. (2018) showed that summer solstice marks a seasonal shift in plant growth, leaf physiology, and foliage turnover in temperate and boreal forests.

In the original version of the CLASS-CTEM model, continuous allocation of carbon to leaves up to the time until they are completely shed led to increase in LAI throughout the growing season rather than a near constant value or slowly decreasing LAI after the summer solstice.

**2.1.4 Adjustments to the lower air temperature threshold**

The CLASS-CTEM model is able to respond to environmental conditions and to transition between different leaf phenological states (Arora and Boer, 2005). Leaf litter ($D_L$) generation is caused by normal turnover of leaves as well as drought and cold stresses which all contribute to LAI seasonality.

$$D_L = C_L \left[ 1 - e^{\left( -\Omega_N - \Omega_C - \Omega_D \right)} \right] \tag{10}$$

where $C_L$ is the leaf carbon pool and $\Omega_{N,C,D}$ are the leaf loss rates (day$^{-1}$) associated with normal (N)

turnover of leaves and the cold (C) and drought (D) stresses. The leaf loss rate associated with cold stress ($\Omega_C$) is based on Eqs. A49-50 of Melton and Arora (2016) (shown below as Eq. 11)

$$\Omega_C = \Omega_{C,max} \, L_{cold}^3 \tag{11}$$

where $\Omega_{C,max}$ is the maximum leaf loss rate due to cold stress and $L_{cold}$ is a scalar that varies between 0

and 1 as follows

$$\quad L_{cold} = \begin{cases} 1 & , T_a < \left( T_{cold}^{leaf} - 5 \right) \\ 1 - \dfrac{T_a - \left( T_{cold}^{leaf} - 5 \right)}{5} & , \left( T_{cold}^{leaf} - 5 \right) < T_a < T_{cold}^{leaf} \\ 0 & , T_{cold}^{leaf} < T_a \end{cases} \qquad (12)$$

$T_{cold}^{leaf}$ is a PFT dependent parameter below which a PFT experiences damage to its leaves and this promotes leaf loss due to cold stress in the model.

The original version of the model used a $T_{cold}^{leaf}$ parameter value of 8 °C throughout the year. In the modified version of the model used here for the broadleaf cold deciduous tree PFT a $T_{cold}^{leaf}$ value of 12

°C is used after summer solstice. For broadleaf cold deciduous tree PFT, leaf out starts in spring, the maximum LAI occurs between July to September (during the northern hemisphere's summer) and the leaves are shed between October and November during autumn. Increasing $T_{cold}^{leaf}$ leads to more leaf litter generation due to the cold stress in the autumn and moves the descending side of the LAI curve inwards during autumn.

**2.2 Model evaluation and experimental set up**

**2.2.1 Description of Fluxnet sites**

We evaluate the performance of the original and modified versions of the CLASS-CTEM

framework in simulating leaf phenology at three well studied sites in the Eastern United States which are selected from the Fluxnet network: (1) Harvard Forest (US-Ha1) located at 42.53 ºN and 72.17 ºW, (2) Morgan Monroe State Forest (US-MMS) at 39.32 ºN and 86.41 ºW, and (3) University of Michigan

Biological Station (US-UMB) at 45.55 ºN and 84.71 ºW. The location of the three Fluxnet sites is shown in Figure 3. The selected sites meet our requirement of availability of observation-based LAI

data (against which our model results can be evaluated) and are primarily characterized by deciduous broadleaf forests although with different species composition. The LAI measurements are based on an LAI-2000 plant canopy analyzer instrument and details are provided in Urbanski et al. (2007), Schmid et al. (2000), and Gough et al. (2008) for the Harvard Forest, Morgan Monroe, and University of Michigan sites, respectively. The mean annual climate at these sites and their species composition are summarized in Table 2. While these sites differ somewhat in the climate they experience, they share enough commonalities in climate to exhibit similar seasonal dynamics of LAI. Annual precipitation at these temperate locations (US-Ha1, US-MMS, and US-UMB) is 1189, 1083, and 613 mm, with an annual mean temperature of 8.2, 12.4, and 7.2 ºC for each site, respectively. These annual averages are based on the half-hourly meteorological data that are used to drive the CLASS-CTEM model for the time period summarized in Table 2.

The US-Ha1 site is owned by Harvard University. Most of its surrounding area was cleared for agriculture during European settlement in 1600-1700. The trees at the site have been regrowing since before 1900 and are currently characterized by predominantly red oak and red maple, with patches of mature hemlock stand and individual white pine. Climate measurements have been made at the Harvard Forest since 1964. The US-MMS site is owned by the Indiana Department of Natural Resources. Many of trees in the tower footprint are 60-80 years old. Today, the forest is a secondary successional broadleaf forest within the maple-beech to oak-hickory transition zone of the eastern deciduous forest. Finally, the US-UMB site is located within a protected forest owned by the University of Michigan and consists of mid-aged northern hardwoods, conifer understory, aspen, and old growth hemlock.

The permeable soil depths are specified at 2.5, 2.5, 2.62 m at the US-Ha1, US-MMS, and US-UMB sites, respectively. Soil texture information was adapted from the global data set of Zobler (1986) and used to specify the percentage of sand and clay in the model's three soil layers as follows. At US-Ha1, the percentages of sand in the first, second and third soil layers are specified at 68.5, 66.5, and 72.25%, and the percentage of clay at 5, 5 and 4.25%, respectively. At US-MMS, the percentages of sand in the first, second and third soil layers are specified at 21, 22.5 and 30.25%, and the percentage of clay at 21, 23 and 23.75%, respectively. At US-UMB, the percentages of sand in the first, second and third soil layers are specified at 71, 72.5 and 73.25%, and the percentage of clay at 7, 7 and 7.75%, respectively.

**2.2.2 CLASS-CTEM simulations**

For the three sites investigated here, we have used version 3.6 of the CLASS coupled to version 2.1.1 of the CTEM model and made changes mentioned above in Section 2.1. Model performance is evaluated for both the modified and original (without NSC pools) versions against available observation-based estimates of LAI, and energy and $CO_2$ fluxes. Simulations were performed for the broadleaf cold deciduous tree PFT with 100% fractional cover.

Seven meteorological variables are required to drive the CLASS-CTEM model - air temperature, air pressure, wind speed, incoming short wave radiation, incoming long wave radiation, precipitation, and specific humidity. Fluxnet's gap-filled meteorological forcing was obtained for each of the three Fluxnet sites. The data were either available at a half-hourly time step or were linearly interpolated from hourly to half-hourly resolution. The meteorological data used to drive the model correspond to the period 1998-2013 for the site in Harvard forest, 1999-2006 for the site in Morgan Monroe State Forest and 1997-2013 for the site at the University of Michigan Biological Station.

All simulations were forced with meteorological data from their respective Fluxnet sites repeatedly until model carbon pools reached equilibrium and the annually averaged NEP was close to zero. A specified atmospheric $CO_2$ concentration of 350 ppm is used at all sites for this spin up. The real world forests have, of course, experienced a gradual increase in atmospheric $CO_2$ concentration, changes in climate, and disturbances over their life time. Although not perfect, in the absence of full histories of disturbance and meteorological data at these sites this approach still allows comparison of the seasonality of simulated LAI and primary carbon and energy fluxes with observation-based estimates once the model pools reach equilibrium. One caveat is that the modelled vegetation and soil carbon pools cannot be expected to be exactly the same as in the real world but we still expect them to be reasonable. We have chosen to use atmospheric $CO_2$ concentration of 350 ppm to spin up the model pools (while the average $CO_2$ concentration during the first decade of the $21^{st}$ century was around 380

ppm) because the terrestrial biosphere is not in equilibrium with the atmospheric $CO_2$ concentration.

The disturbance (fire) module was not activated in these simulations. Observation-based LAI

measurements were obtained from the Ameriflux web site (*https://ameriflux.lbl.gov*). Energy and $CO_2$

fluxes were obtained from the Fluxnet web site (*https://fluxnet.fluxdata.org*).

**3 Results**

Model performance is evaluated by comparing simulated LAI and $CO_2$ fluxes of gross primary productivity (GPP) and net ecosystem productivity (NEP) which is our primary focus. We also compare radiative energy fluxes of net radiation and latent and sensible heat with their observation- based estimates from the modified and the original model versions.

**3.1 LAI and land-atmosphere $CO_2$ fluxes**

Figures 4-6 compare simulated values of LAI, GPP, NEP, and $E_r$ from the two model versions with their observation-based estimates at the US-Ha1, US-MMS, and US-UMB Fluxnet sites. Observation- based measurements are shown in black and simulated mean daily values are shown in red (for the original model version indicated as CLASS-CTEM Original) and blue (for the modified version, with

NSC pools and other changes indicated as CLASS-CTEM Modified). Just like simulated values, the observation-based estimates also represent average daily values across all years for which the data were available. The mean annual values of LAI, GPP, $E_r$, and NEP are also summarized in Table 3. At all sites, when compared to the original version, the modified version of the model shows a phenological shift of about 2 weeks earlier in the year which is in better agreement with observed LAI transitions (Figs. 4-6, panel a). The timing of maximum LAI also improves and shows a shift of about 2 months earlier in the year, from late September and early October to late July and early August. The observation-based estimates of LAI suggest the presence of understory vegetation at two of the three

Fluxnet sites (the Monroe and the Michigan sites). The CLASS-CTEM modelling framework does not represent any understory vegetation. Despite this, the model still overestimates maximum LAI at all locations and its implications are discussed further down. At all three sites, the inclusion of non- structural carbon pools (section 2.1.1) and other model modifications (sections 2.1.2 to 2.1.4) produces a notable improvement in simulated LAI seasonality, especially during canopy development (i.e., spring and early summer) and its autumn decline.

The Morgan Monroe site (Fig. 5) experiences somewhat warmer temperatures than the Harvard and

Michigan sites (Figs. 4 and 6) (mean annual temperature at the Morgan Monroe is about 4 ºC higher than at the other two sites, see Table 2) and as a result the growing season is somewhat longer at the

Morgan Monroe site. The model is able to successfully capture this difference amongst the sites.

Overall, the simulated GPP and NEP (Figs. 4-6, panels b and c) compare reasonably well with observations. Improvements in simulated LAI seasonality lead to concomitant improvements in simulated GPP especially at the ascending side of the plots when the growing season starts. In the original version of the model the increase in GPP at the start of the growing season is delayed due to delayed leaf out. Note that the simulated GPP values compare well with their observation-based estimates despite the higher simulated LAI. Improvements in simulated GPP also lead to improvements in simulated NEP in Figures 4-6 (panel c), and similar to GPP, especially on the ascending side of the plots at the start of the growing season.

The comparison with observation-based estimates of LAI and GPP is not completely straightforward since the observation-based estimates of these two quantities are put together by different communities and the fact that GPP is a derived quantity (as opposed to NEP which is directly observed). As a result, the observation-based estimates of LAI and GPP are not completely consistent with each other. This is seen in Figure 5 for the US-MMS site where the modified version of the model results in a better match with observation-based estimate of LAI (Fig. 5, panel a), but it shows a bias towards an early increase in GPP (Fig. 5, panel b). In contrast, in Figure 6, the simulated GPP in the modified version of the model compares better with its observation-based estimate than the LAI. In this respect, NEP provides a better measure to assess model improvement than GPP. In Figures 4 to 6

improvements in simulated LAI in the modified version of the model are more consistent with improvements in simulated NEP.

The individual contributions of the three modifications, 1) inclusion of NSC pools, 2) reduced allocation of carbon to leaves after summer solstice, and 3) change in parameter value of temperature threshold for leaf litter loss due to cold stress, made to the model for resulting improvements in LAI,

GPP, and NEP are shown in the supplementary information.

The annual mean of observation-based NEP values (as shown in the figure legends of Figures 4 to

6) is positive because northern hemisphere temperate land is currently a sink of carbon (Myneni et al.,

2001). In contrast, the annual mean of simulated NEP values is close to zero by construction, because the model was spun-up to an equilibrium state. The positive annual mean of observation-based NEP

values, compared to simulated NEP values, can manifest in multiple ways – as primarily higher summer values when NEP values are positive (as for the Harvard forest site), as higher values through the year (as is mostly the case at the Morgan Monroe site) and as less negative NEP values during non-growing season when NEP values are negative (as seen at the University of Michigan site). Regardless of this caveat, the inclusion of NSC pools to advance leaf onset and offset times does lead to an improvement in seasonality of simulated NEP values.

While photosynthesis primarily depends on the current meteorological conditions and LAI amongst other environmental factors (including atmospheric $CO_2$ concentration), ecosystem respiration (Figs. 4-6, panel d) depends strongly on the vegetation and soil carbon pool sizes. As a result, if simulated vegetation and soil carbon pools are larger or smaller than observation-based estimates then so would be the respiratory fluxes. Note also that the simulated annual respiratory fluxes are higher than observed at all three sites (Figs. 4-6, panel d, and Table 3). Had the simulated fluxes been lower than what they are now and closer to their observation-based estimates, then the simulated NEP would have been more similar to observations. Nevertheless, the model simulates the seasonality of ecosystem respiratory fluxes reasonably well. In absence of the long term disturbance history or meteorological data to drive the model with, the current methodology (where the model is driven repeatedly with the available observed meteorological data) is reasonable and allows us to assess the seasonality of simulated LAI and land-atmosphere $CO_2$ fluxes - which is the primary objective of our study.

**3.2 NSC pools**

Figures 7-9 evaluate the seasonal cycle of the NSC pools in leaf, stem, and root vegetation components at US-Ha1, US-MMS, and US-UMB Fluxnet sites, respectively. There are no observation-based estimates of NSC pools available at the three Fluxnet sites. For the broadleaf cold deciduous tree PFT considered here, the stem carbon pool is the largest (and so are its structural and non-structural parts) and the leaf carbon pool is the smallest (Figs. 7-9, panels a and c). The amount of non-structural carbon reallocated from the stem and root NSC pools to leaves during leaf onset in early spring (see section 2.1.2) is shown in Figures 7-9 (panel b). Figures 7-9 (panel d) show the seasonality of the carbon flux from the non-structural to the structural part of the leaf, stem and root components for the three sites. The seasonality of total stem and root carbon pools is driven mostly by the seasonality of their non-structural parts.

For the stem and root components, the non-structural parts contribute about 6-10% to the total pool size. During the early leaf-out period when reallocation from stem and root NSC pools to leaves is taking place (section 2.1.2), the stem's NSC pool gets depleted. This transfer/reallocation stops after a threshold LAI is achieved. The transfer of NSC from stem and root pools to leaves occurs mostly through the stem (see Figs. 7-9, panel b) since its NSC pool is about 3-4 times larger than the root component. The NSC pool for both components reduces during the period when leaves are not present (and GPP is zero) due to respiratory and litter losses. The pools for both stem and root components get replenished later during the growing season when a sufficient amount of leaves has been grown and allocation of carbon to stem and root components is restored. This is seen in Figures 7-9 (panel d)

which show the flux of carbon from non-structural to structural leaf, stem and root components. Early on during the growing season, carbon flux from the leaf NSC pool to its structural part is much higher since the model preferably allocates carbon to leaves as discussed in section 2.1.2. After a threshold

LAI is reached, carbon is also allocated to stem and root NSC pools which subsequently start to allocate carbon to their structural pools and the tree biomass continues to increase. At the end of the growing season, when photosynthesis stops, allocation to all three components and the fluxes from

NSC to structural parts terminate. During the dormant winter season NSC pools provide for the respiratory costs.

**3.3 Energy fluxes**

Figure 10 compares observation-based measurements of latent heat (LE), sensible heat (H), and net radiation ($R_n$) fluxes at the three Fluxnet sites, with their simulated values from the two model versions.

Annual mean values of these observation-based and simulated radiative and turbulent energy fluxes are also summarized in Table 3. Unlike the simulated fluxes, the annual mean sum of the observed LE and

H, averaged over the years for which observations are available, is not equal to the observed $R_n$. This non-closure of the energy budget is seen at all three sites and is a typical characteristic of eddy covariance based flux measurements (Gao et al., 2017). The annual energy budget closure is off by

17% at the University of Michigan Biological Station, by 20% at the Harvard Forest and by 30% at the

Morgan Monroe sites as seen in Table 3. Keeping this caveat in mind,  the model overall captures the seasonality of radiative and turbulent fluxes shown in Fig. 10 reasonably, with the exception of late winter and early spring. During this period, as solar radiation increases $R_n$ is underestimated (Fig. 10, panels a-c) until the canopy approaches a full-leaf state and this leads to an underestimation of H (Fig.

10, panels g-i) and overestimation of LE. This may be caused by an overestimation of canopy transmissivity and underestimation of snow and soil masking by leafless forests with increasing solar elevation (recently observed in unpublished simulations with CLASS-CTEM at the Borden forest,

Borden, Ontario, Canada), and may also be exacerbated by the lack of representation of a small evergreen needleleaf fraction at US-Hal and a conifer understory at US-UMB. LE is apparently overestimated throughout the year at US-MMS but we suspect this reflects a larger underestimation of

LE relative to H in the measured fluxes; Oliphant et al. (2004) found that accounting for long sampling tube damping effects on LE resulted in a 16% improvement in energy balance closure at this site. The change to an earlier leaf phenology in the modified simulations results in a slightly earlier increase in

LE in the spring, as well as slightly earlier decreases in autumn at US-Hal and US-UMB, but differences are much smaller at US-MMS.

In Figure 10 the changes in LAI, due to modifications made to the original version of the model, do not significantly affect latent heat fluxes because of two related reasons. First, at mid-high latitude locations where soil moisture constraint is not very large, as is the case at the three sites considered in this study, total evapotranspiration (or latent heat flux) is controlled by available energy. This is the reason for the expected seasonality in latent heat flux at these sites which is characterized by higher values during summer and lower values during winter. Second, since the latent heat flux at these three sites is controlled primarily by available energy, the resulting implication is that if evaporative demand cannot be met by transpiration then it will be met by evaporation from the soil. As a result, changes in LAI do not significantly affect total evapotranspiration but change the partitioning of evapotranspiration flux coming from transpiration, evaporation of intercepted water on canopy leaves, and evaporation from the soil.

**4. Discussion and conclusions**

The CLASS-CTEM model, similar to other land surface schemes implemented in other Earth system models, is not tuned for any specific location but is expected to behave realistically at all locations. Model processes correspond to generic PFTs, in this case broadleaf cold deciduous trees, and are not meant to represent specific species differences within a PFT. It is nearly impossible, at present, to determine model's the more than 100 parameters for individual species. As a result, while our three chosen sites are characterized by different species (as shown in Table 2) they must be represented by a single set of parameter values which correspond to the broadleaf cold deciduous PFT.

Previous studies using the CLASS-CTEM model in the context of land-atmosphere $CO_2$ fluxes and simulated carbon pools have evaluated its performance at point (Arora, 2003; Arora and Boer, 2005;

Melton et al., 2015), regional (Garnaud et al., 2015; Peng et al., 2014; Arora et al., 2016) and global (Arora and Boer, 2010; Melton and Arora, 2014, 2016) scales. These studies indicate that the model performance is reasonable. CLASS-CTEM also participated in the TRENDY model intercomparison, the result of which contributed to the Global Carbon project for years 2016 and 2017 (Le Quéré et al., 2016, 2017). A typical model evaluation exercise at the global and regional scale compares model-simulated geographical distribution of GPP, vegetation biomass, and soil carbon with their respective observation-based estimates. Point scale studies, on the other hand, typically focus on the simulated seasonality of energy and $CO_2$ fluxes as is the case in this study. Model evaluation exercises not only help in identifying model limitations but also yield opportunities to improve model performance by tuning model parameters.

We chose to perform equilibrium simulations by forcing the model repeatedly with available meteorological data at specified $CO_2$ concentration of 350 ppm. In the absence of meteorological data that shows a warming trend over the historical period we would not have been able to properly perform a historical simulation. Our past experience shows that steady state simulations, as opposed to historical transient simulations, allow an easier interpretation of model modifications in the absence of confounding effects of changing climate and increasing $CO_2$.

Previous evaluations of the CLASS-CTEM model that highlighted its limitation of delayed leaf phenology (e.g., Anav et al., 2013) were the motivation for this study. NSC pools play an important role during leaf onset for broadleaf deciduous cold trees, but also other PFTs, and their effect in the original model was represented using the concept of imaginary leaves whose LAI is assumed to be directly proportional to non-leaf biomass. Here, we have included NSC pools in the model framework explicitly along with some other changes and these modifications do lead to improvement in simulated leaf phenology and concomitant improvements in simulated seasonal cycle of GPP and NEP.

Improvements in simulated energy fluxes are much harder to detect because the observation-based energy fluxes are affected by non-closure of the energy budget but also because latent heat fluxes are not as strongly dependent on LAI as GPP.

Despite the simulated LAI being higher than observation-based estimates the simulated GPP, $E_r$, and NEP compare reasonably with their observation-based estimates. Possible reasons for higher simulated LAI include higher than observed allocation to leaf compared to stem and root components and lower than observed leaf turnover and/or leaf respiration rates. The model currently uses a maximum carboxylation capacity ($V_{c,max}$, i.e., maximum photosynthetic rate) value of 57 u-mol $CO_2$/

$m^2s$ for broadleaf cold deciduous trees based on Table 3 of Kattge et al. (2009) who derive $V_{c,max}$ values for major PFTs using more than 700 data estimates. While in the CLASS-CTEM model photosynthesis is also limited by light and transport capacity rates in addition to carboxylation capacity (see Appendix

A2 in Melton and Arora, 2016), $V_{c,max}$ remains a strong parameter and simulated GPP in the model is proportional to $V_{c,max}$. While the model simulated LAI can be lowered by tuning allocation to leaves, leaf turnover and/or respiration rates specifically for these sites, this would imply using a $V_{c,max}$ value higher than that suggested by Kattge et al. (2009) to achieve realistic GPP. It is possible that the average $V_{c,max}$ value derived by Kattge et al. (2009) is not representative of broadleaf cold deciduous trees in the Eastern United States. The simulated LAI in the model is the result of multiple model processes interacting with each other. We note this limitation of the model at these locations and plan to address it in near future. While LAI is an important determinant of model performance even more important are the land-atmosphere $CO_2$ fluxes from an ESM perspective since it is the $CO_2$ fluxes which determine the carbon budget of the atmosphere in a fully coupled ESM simulation (Arora et al.,

2013).

Plants are extremely complex living organisms which respond to the changes in their physical and chemical environmental conditions using a myriad of adaptations. Our limited understanding of these adaptations comes only from empirical observations of their behaviour and measurement of their physical and chemical responses to environmental changes. Models typically represent only a fraction of this understanding because model structures depend on the purpose of the model and the amount of details that can be represented reasonably in a model's framework. In hindsight, the omission of NSC pools in the original model version was a structural error and while the conceptual imaginary leaves tried to mimic the fast growth rate of leaves during leaf onset at the arrival of favourable environmental conditions they were not completely successful in capturing the real-world behaviour. In addition, in a past exercise, we also used higher $V_{c,max}$ values at the beginning of the growing season to accelerate the rate of growth of leaves (based on Bauerle et al. (2012) and Alton (2017)) but this also did not help sufficiently to address the slower than observed rate of growth of LAI at the start of the growing season. Unlike physical models, which describe a physical process, modelling of biological response to changes in environmental conditions is more complex. While there may be underlying physical laws that determine the response of plants to changes in environmental conditions, we can only interpret this with a biological point of view. Dynamic vegetation models and land surface schemes parameterize biological functioning using mathematical formulations to reproduce empirical observations and modellers' conceptual understanding of how the biology works. The inclusion of NSC pools in the CLASS-CTEM framework is based on the same philosophy.

The implementation of NSC pools in the CLASS-CTEM modelling framework presented in this study is meant specifically to address the problem of delayed leaf phenology. NSC pools also play a vital role in the overall health of the plants as mentioned earlier in the introductory section. During periods of limited photosynthesis, trees depend solely on stored NSCs to maintain basic metabolic functions, produce defensive compounds, and retain cell turgor (Sperling et al., 2015). A period of continuous drought, for instance, will gradually reduce the size of NSC pools and this can be used as a trigger to initiate drought related mortality in the model, or alternatively NSC pools may be used to allow leaf growth during a short-term dry period to represent resilience (Mitchell et al., 2013). The inclusion of NSC pools also lays the groundwork to implement a nitrogen (N) cycle in the CLASS-

CTEM framework since modelling $V_{c,max}$ as a function of leaf N content requires leaf N content in the non-structural part of the leaves.

In conclusion, modifications to the CLASS-CTEM framework made in this study to address the problem of delayed leaf phenology yield improvements to simulated seasonality of LAI at the three

Fluxnet sites considered here. These improvements, especially the inclusion of NSC pools also lay the groundwork for future model development and inclusion of new processes.

*Acknowledgements*

A. Asaadi was supported by a National Scientific and Engineering Research Council of Canada (NSERC) Visiting Postdoctoral Fellowship. We thank Fluxnet and AmeriFlux data networks for providing the data used in our study. We would also like to thank Philip Savoy for sharing LAI data for the Morgan Monroe site investigated in this study. We are also grateful to Reinel Sospedra-Alfonso and

Michael Sigmond for providing comments on an earlier version of this manuscript.

**List of Tables**

Table 1: Plant functional types (PFTs) represented in CTEM and their relation to CLASS PFTs.

| CLASS PFTs | CTEM PFTs |
|---|---|
| Needleleaf trees | Needleleaf Evergreen trees |
| | Needleleaf Deciduous trees |
| Broadleaf trees | Broadleaf Evergreen trees |
| | Broadleaf Cold Deciduous trees |
| | Broadleaf Drought/Dry Deciduous trees |
| Crops | C3 Crops |
| | C4 Crops |
| Grasses | C3 Grasses |
| | C4 Grasses |

Table 2. The location of Fluxnet sites, primary species that exist at these sites, their soil physical characteristics, mean annual values of primary meteorological variables, and years of data availability

| Site Name | Harvard forest (US-Ha1) | Morgan Monroe state forest (US-MMS) | Uni. of Mich. Biological station (US-UMB) |
|---|---|---|---|
| Lat, Lon,Elevation | 42.53º, -72.17º, 340m | 39.32º, -86.41º, 275m | 45.55º, -84.71º, 234m |
| Biome Type | Broadleaf deciduous forest | Broadleaf deciduous forest | Broadleaf deciduous forest |
| Species | Red Oak (Quercus rubra), Red Maple (Acer rubrum), Hemlock (Tsuga canadensis), White Pine (Pinus strobus) | Maple-beech (Fagus grandifolia), Oak-Hickory | Conifer understory, Aspen (Populus tremuloides), Hemlock (Cicuta), and other northern hardwood trees |
| Mean annual air T (ºC) | 8.2 | 12.4 | 7.2 |
| Mean annual precip. (mm) | 1189 | 1083 | 613 |
| Mean annual SW Radiation (W/m²) | 151 | 167 | 154 |
| Mean annual LW Radiation (W/m²) | 263 | 329 | 299 |
| Soil depth (m) | 2.5 | 2.5 | 2.6 |
| % of soil sand (layer 1, 2, 3) | 68.5, 66.5, 72.25 | 21, 22.5, 30.25 | 71, 72.5, 73.25 |
| % of soil clay (layer 1, 2, 3) | 5, 5, 4.25 | 21, 23, 23.75 | 7, 7, 7.75 |
| Years for which LAI data are available | 1998-2013 | 1999-2006 | 1997-2013 |

| Site name | | Harvard forest (US-Ha1) | Morgan Monroe (US-MMS) | Uni. of Mich. (US-UMB) |
|---|---|---|---|---|
| Land -atmosphere $CO_2$ fluxes (gC m$^{-2}$ yr$^{-1}$) and LAI (m$^2$/m$^2$) | | | | |
| Gross primary productivity | Observed | 3.9 | 4.5 | 3.6 |
| | CLASS-CTEM original | 3.6 | 5.0 | 3.6 |
| | CLASS-CTEM modified | 3.7 | 5.3 | 3.7 |
| Ecosystem respiration | Observed | 3.3 | 3.3 | 2.9 |
| | CLASS-CTEM original | 3.6 | 4.9 | 3.5 |
| | CLASS-CTEM modified | 3.7 | 5.3 | 3.7 |
| Net ecosystem productivity | Observed | 0.7 | 1.2 | 0.7 |
| | CLASS-CTEM original | 0.02 | 0.1 | 0.0 |
| | CLASS-CTEM modified | 0.0 | 0.0 | 0.0 |
| Leaf area index | Observed | 1.8 | 3.0 | 2.8 |
| | CLASS-CTEM original | 2.0 | 3.1 | 1.9 |
| | CLASS-CTEM modified | 1.9 | 3.0 | 1.8 |
| Energy fluxes and energy budget (W/m$^2$) | | | | |
| Net radiation ($R_n$) | observed | 78.9 | 89.6 | 78.1 |
| | CLASS-CTEM original | 59.6 | 89.2 | 66.8 |
| | CLASS-CTEM modified | 59.6 | 89.6 | 67.2 |
| Latent heat flux (LE) | observed | 34.0 | 38.3 | 35.4 |
| | CLASS-CTEM original | 38.9 | 68.6 | 47.9 |
| | CLASS-CTEM modified | 39.4 | 69.3 | 48.3 |
| Sensible heat flux (H) | observed | 29.3 | 25.1 | 29.4 |
| | CLASS-CTEM original | 20.7 | 20.6 | 18.9 |
| | CLASS-CTEM modified | 20.2 | 20.3 | 18.9 |
| $R_n$-LE-H | observed | 15.6 | 26.2 | 13.3 |
| | CLASS-CTEM original | 0 | 0 | 0 |
| | CLASS-CTEM modified | 0 | 0 | 0 |

**List of Figures**

CLASS-CTEM (modified version) simulated values of total (panel a) and -non-structural (panel c) carbohydarte pools (Kg C/m$^2$). Panel (b) shows the reallocation of carbon from non-structural stem and root pools to leaves during leaf onset in spring and panel (d) shows the carbon flux from non-structural to structural pools for leaf, stem and root components (gC/m$^2$.day) for US-Ha1(Harvard Forest) Fluxnet site. The plots show mean daily values across all years for which the meteorological data were available after the model pools reached equilibrium.

CLASS-CTEM (modified version) simulated values of total (panel a) and  non-structural (panel c) carbohydarte pools (Kg C/m$^2$). Panel (b) shows the reallocation of carbon from non-structural stem and root pools to leaves during leaf onset in spring and panel (d) shows the carbon flux from non-structural to structural pools for leaf, stem and root components (g.C/m$^2$.day) for US-MMS (Morgan Monroe State Forest) Fluxnet site. The plots show mean daily values across all years for which the meteorological data were available after the model pools reached equilibrium.

CLASS-CTEM (modified version) simulated values of total (panel a) and  non-structural (panel c) carbohydarte pools (Kg C/m$^2$). Panel (b) shows the reallocation of carbon from non-structural stem and root pools to leaves during leaf onset in spring and panel (d) shows the carbon flux from non-structural to structural pools for leaf, stem and root components (g.C/m$^2$.day) for US-UMB (University of Michigan Biological Reserve) Fluxnet site. The plots show mean daily values across all years for which the meteorological data were available after the model pools reached equilibrium.

Observed and CLASS-CTEM simulated daily net radiation (W/m$^2$), latent heat flux (W/m$^2$), and sensible heat flux (W/m$^2$) for the three Fluxnet sites. Legends also show the mean annual value of the quantity plotted. Root mean square error (RMSE) and coefficient of determination ($R^2$) are also shown for simulated values when compared to observation-based estimates.

[Figure]

Figure 1: Schematic representation of the CTEM model after addition of non-structural carbohydrate pools. The arrows in blue color show the new non-structural carbohydrate fluxes as shown in Equations 5 and 6.

[Figure]

Figure 2: Latitude dependence factor ($\Gamma$) (Equation 8) for reducing allocation fraction to leaves after summer solstice.

[Figure]

Figure 3: Location of the three Fluxnet sites chosen in this study to evaluate the changes made to the CLASS-CTEM parameterizations aimed to improve leaf phenology. Figure adapted from Google maps.

[Figure]

Figure 4: Observed and CLASS-CTEM simulated averaged daily values of a) LAI ($m^2/m^2$), b) GPP (g.C/$m^2$.day), c) NEP (g.C/$m^2$.day), and d) Ecosystem respiration (g.C/$m^2$.day) for US-Ha1 (Harvard Forest) Fluxnet site across all years were data are present. Legends also show the mean annual value of the quantity plotted, except for LAI which is averaged over the growing season. Root mean square error (RMSE) and coefficient of determination ($R^2$) are also shown for simulated values when compared to observation-based estimates.

[Figure]

Figure 5: Observed and CLASS-CTEM simulated averaged daily values of a) LAI (m$^2$/m$^2$), b) GPP (g.C/m$^2$.day), c) NEP (g.C/m$^2$.day), and d) Ecosystem respiration (g.C/m$^2$.day) for US-MMS (Morgan Monroe State Forest) Fluxnet site across all years were data are present. Legends also show the mean annual value of the quantity plotted, except for LAI which is averaged over the growing season. Root mean square error (RMSE) and coefficient of determination (R$^2$) are also shown for simulated values when compared to observation-based estimates.

[Figure]

Figure 6: Observed and CLASS-CTEM simulated averaged daily values of a) LAI ($m^2/m^2$), b) GPP ($g.C/m^2.day$), c) NEP ($g.C/m^2.day$), and d) Ecosystem respiration ($g.C/m^2.day$) for US-UMB (University of Michigan Biological Reserve) Fluxnet site across all years were data are present. Legends also show the mean annual value of the quantity plotted, except for LAI which is averaged over the growing season. Root mean square error (RMSE) and coefficient of determination ($R^2$) are also shown for simulated values when compared to observation-based estimates.

[Figure]

Figure 7: CLASS-CTEM (modified version) simulated values of total (panel a) and non-structural (panel c) carbohydarte pools (Kg C/m$^2$). Panel (b) shows the reallocation of carbon from non-structural stem and root pools to leaves during leaf onset in spring and panel (d) shows the carbon flux from non-structural to structural pools for leaf, stem and root components (gC/m$^2$.day) for US-Ha1(Harvard Forest) Fluxnet site. The plots show mean daily values across all years for which the meteorological data were available after the model pools reached equilibrium.

[Figure]

Figure 8: CLASS-CTEM (modified version) simulated values of total (panel a) and non-structural (panel c) carbohydarte pools (Kg C/m$^2$). Panel (b) shows the reallocation of carbon from non-structural stem and root pools to leaves during leaf onset in spring and panel (d) shows the carbon flux from non-structural to structural pools for leaf, stem and root components (g.C/m$^2$.day) for US-MMS (Morgan Monroe State Forest) Fluxnet site. The plots show mean daily values across all years for which the meteorological data were available after the model pools reached equilibrium.

[Figure]

Figure 9: CLASS-CTEM (modified version) simulated values of total (panel a) and non-structural (panel c) carbohydarte pools (Kg C/m$^2$). Panel (b) shows the reallocation of carbon from non-structural stem and root pools to leaves during leaf onset in spring and panel (d) shows the carbon flux from non-structural to structural pools for leaf, stem and root components (g.C/m$^2$.day) for US-UMB (University of Michigan Biological Reserve) Fluxnet site. The plots show mean daily values across all years for which the meteorological data were available after the model pools reached equilibrium.

[Figure]

Figure 10: Observed and CLASS-CTEM simulated daily net radiation (W/m²), latent heat flux (W/m²), and sensible heat flux (W/m²) for the three Fluxnet sites. Legends also show the mean annual value of the quantity plotted. Root mean square error (RMSE) and coefficient of determination (R²) are also shown for simulated values when compared to observation-based estimates.